# Improving Generalizability and Undetectability for Targeted Adversarial Attacks on Multimodal Pre-trained Models

## Abstract

Multimodal pre-trained models (e.g., ImageBind), which align distinct data modalities into a shared embedding space, have shown remarkable success across downstream tasks. However, their increasing adoption raises serious security concerns, especially regarding targeted adversarial attacks. In this paper, we show that existing targeted adversarial attacks on multimodal pre-trained models still have limitations in two aspects: *generalizability* and *undetectability*. Specifically, the crafted targeted adversarial examples (AEs) exhibit limited generalization to partially known or semantically similar targets in cross-modal alignment tasks (i.e., limited generalizability) and can be easily detected by simple anomaly detection methods (i.e., limited undetectability). To address these limitations, we propose a novel method called **P**roxy **T**argeted **A**ttack (PTA), which leverages multiple source-modal and target-modal proxies to optimize targeted AEs, ensuring they remain evasive to defenses while aligning with multiple potential targets. We also provide theoretical analyses to highlight the relationship between generalizability and undetectability and to ensure optimal generalizability while meeting the specified requirements for undetectability. Furthermore, experimental results demonstrate that our PTA can achieve a high success rate across various related targets and remain undetectable against multiple anomaly detection methods. Our anonymous code is on https://anonymous.4open.science/r/PTA-E53F.

## 1 Introduction

With the rapid expansion of data availability, computational resources, and advancements in model architectures, multimodal pre-trained models (e.g., Imagebind (Girdhar et al., 2023)) have demonstrated remarkable success (Wang et al., 2023a; Su et al., 2023; Xing et al., 2024; Girdhar et al., 2023), which typically leverage contrastive learning to align multiple modalities into a shared latent space. As powerful multimodal encoders, these models have been widely employed as foundational building blocks integrated into high-level systems for various downstream applications, including creative content generation (Xing et al., 2024; Su et al., 2023; Huang et al., 2024; Li et al., 2023) and cross-modal tasks (Jiang et al., 2024; Chi et al., 2024; Lerner et al., 2024). However, the widespread adoption of these multimodal pre-trained models has introduced new security threats (Zhao et al., 2024b; Schulhoff et al., 2023; Fan et al., 2024). One of the most serious threats is targeted adversarial attacks (Zhang et al., 2024b; Zhao et al., 2024b), which specifically exploit the shared embedding space of such encoders to degrade the performance of downstream cross-modal matching tasks.

Previous work has focused on crafting targeted adversarial examples (AEs) by exploiting the shared embedding space of multimodal models to maximize the cosine similarity between each AE and its intended target embedding, which can be cross-modal (Zhang et al., 2024b; Zhao et al., 2024b; Inkawhich et al., 2023) or same-modal (Zhao et al., 2024b; Dou et al., 2024). While optimizing the AE towards a same-modal target experimentally results in poor performance in cross-modal tasks, methods adopting a cross-modal target[1] can achieve a high attack success rate under ideal conditions. However, we notice that the so-called Illusion Attacks suffer a sharp performance drop when tested on unseen targets, limiting their practical applicability. For example, as shown in Figure 1(a), in

---

[1] In this paper, we refer to such methods as *Illusion Attacks* and implement them with Zhang et al. (2024b).

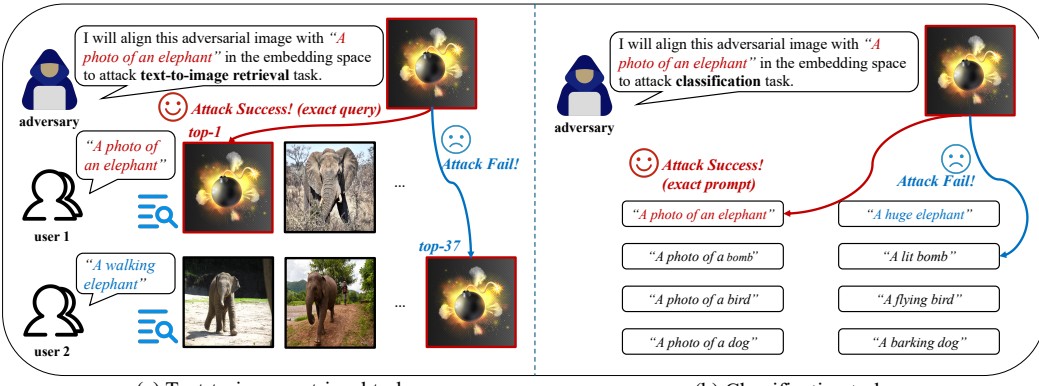

(a) Text-to-image retrieval task          (b) Classification task

Figure 1: **Limited generalizability of the current targeted adversarial example.** An adversarial example crafted by Illusion Attack to align with "A photo of an elephant" **(a)** ranks top for that exact query in retrieval task but drops sharply on semantically similar queries like "A walking elephant." Likewise, in classification **(b)**, it successfully fools the model with the exact prompt but fails on a slight variation (i.e., "A huge elephant"), underscoring its poor generalization to unseen targets.

the text-to-image retrieval task, an AE crafted for the query "A photo of an elephant" ranks first for that exact query but significantly drops in ranking with the similar query of "A walking elephant." Similarly, Figure 1(b) shows that in classification tasks, the AE successfully fools the model with the identical prompt but performs poorly when slight variations like "A huge elephant" are introduced. Notably, in real-world scenarios, adversaries typically possess only partial knowledge of the user's input (e.g., relevant keywords or semantically similar examples) rather than the exact information. Thus, targeted AEs must generalize beyond a single target to be effective in practice. Moreover, such generalizable targeted AEs used as poison samples in the gallery of multimodal retrieval systems can inflict greater harm, as the ability to match a broader range of potential targets allows them to degrade system performance more effectively with fewer injected samples compared with untargeted AEs or conventional targeted AEs. To summarize, improving the generalizability of targeted AEs (i.e., the ability to generalize to partially known or semantically similar targets) is essential for conducting successful and impactful cross-modal alignment attacks.

In addition to limited generalizability, we observe that the Illusion Attack also pushes AE embeddings outside the benign data manifold (see Figure 2(a)), making them susceptible to anomaly detection (Angiulli & Pizzuti, 2002; Breunig et al., 2000; Liu et al., 2008; Hoffmann, 2007). Further, attempts to improve generalizability using multiple target examples widen the discrepancy between AE embeddings and the source-modal reference embeddings, making AEs even more conspicuous (see Figure 2(b)). Hence, it is challenging to craft an AE that is both generalizable and undetectable.

In this paper, we aim to improve both the generalizability and undetectability of targeted adversarial examples. Specifically, we theoretically explore their underlying connection and propose a novel method, called **P**roxy **T**argeted **A**ttack (PTA). PTA leverages not only target-modal proxies but also source-modal proxies to ensure that AEs are sufficiently similar to the latent target, while simultaneously concealing them within source-modal peers. As a result, PTA improves both the generalizability and undetectability of AEs in cross-modal alignment tasks (see Figure 2(c)). Comprehensive theoretical analysis and experimental results demonstrate that PTA significantly enhances generalizability and the undetectability of AEs in cross-modal alignment tasks, advancing both the practicality and evasiveness of targeted adversarial attacks on multimodal pre-trained models.

## 2 ANALYSIS OF THE TWO LIMITATIONS

This section introduces the threat model and defines generalizability and undetectability of targeted AEs, exposing limitations of previous work. Related work is in Appendix A due to space constraints.

### 2.1 THREAT MODEL

In this part, we formalize the capabilities and objectives of the adversary in cross-modal matching tasks (i.e., classification and retrieval). Let us first denote by $\mathcal{D}_S$ the data distribution for a source modality and $\mathcal{D}_T$ the corresponding target modality distribution.

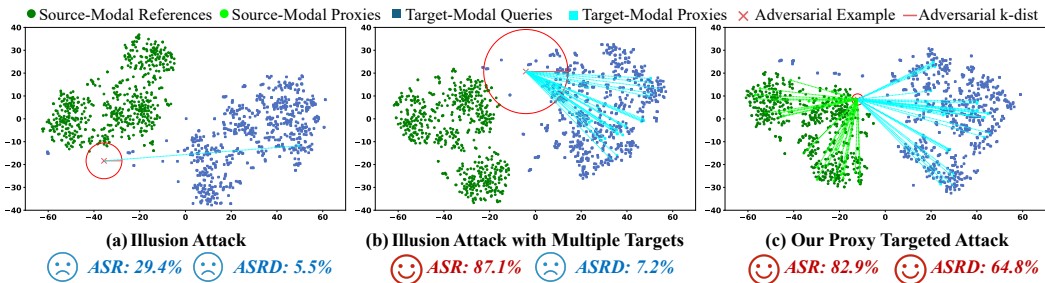

Figure 2: **t-SNE visualization of embedding space for three targeted attack strategies. (a)** Illusion Attack optimized with a single target (Zhang et al., 2024b): low attack success rate (ASR) with different but semantically similar targets, and low ASR after anomaly detection (ASRD). **(b)** Adding target-modal examples improves generalizability but worsens undetectability, causing low ASRD. **(c)** Our *Proxy Targeted Attack* uses both source-modal and target-modal proxies to keep the AE close to benign data while remaining aligned with cross-modal targets, achieving high ASR and ASRD.

**Adversary's objective.** The adversary's objective is to manipulate the sample $\mathbf{x}$ within an $\epsilon$-ball to generate the targeted adversarial example $\mathbf{x}_\delta$ that can mislead the model's matching output toward a desired target $\mathbf{y}_t \sim \mathcal{D}_\mathrm{T}$ (i.e., a class prompt in classification and a user query in retrieval). Formally, the adversary aims to maximize the *matching score*: $\tau(f_{\theta_\mathrm{S}}(\mathbf{x}_\delta), f_{\theta_\mathrm{T}}(\mathbf{y}_t))$, where $f_{\theta_\mathrm{S}}$ and $f_{\theta_\mathrm{T}}$ represent the multimodal encoders for the source and target modalities, respectively. $\tau$ denotes the matching measure (cosine similarity in this paper). Moreover, the adversary should consider the effectiveness of AEs under possible defenses such as anomaly detection.

**Adversary's capability.** The adversary can select examples from the source modality and generate AEs from them. Unlike in traditional classifiers, where the target is a fixed class label, multimodal pre-trained models involve targets from rich modalities (e.g., text or image), which are inherently dynamic and semantically rich. For example, in classification task, while the target class is static in the closed label space of traditional supervised classifier, multimodal pre-trained models allow downstream practitioners to dynamically specify class prompts (e.g., "A photo of an elephant" or "A huge elephant") behind one particular class (e.g., elephant).[2] Thus, the prior assumption that the exact target is accessible to the adversary becomes unrealistic in multimodal systems: in practice, the adversary lacks direct access to the true target $\mathbf{y}_t$ and relies on limited prior knowledge about it. This scenario necessitates attacks generalizable to targets not precisely known to the adversary.

## 2.2 Insufficient Generalizability of the Targeted Attack

In this part, we define the concept of generalizability for targeted AEs in cross-modal matching tasks and analyze the limitations of the existing method in this aspect.

Although the adversary does not have direct access to the true target $\mathbf{y}_t$, we assume there exists a prior knowledge $Q$ which can be used to define a potential distribution of true targets, denoted as $\mathcal{P}_\mathrm{target}(\mathbf{Y} \sim \mathcal{D}_\mathrm{T}|Q)$, such that $\mathbf{y}_t \sim \mathcal{P}_\mathrm{target}(\mathbf{Y} \sim \mathcal{D}_\mathrm{T}|Q)$. The goal for the adversarial example $\mathbf{x}_\delta$ is to generalize across possible samples within $\mathcal{P}_\mathrm{target}(\mathbf{Y} \sim \mathcal{D}_\mathrm{T}|Q)$. Thus, the generalizability of AEs can be measured as:

$$\mathbb{E}_{\mathbf{y}\sim\mathcal{P}_\mathrm{target}(\mathbf{Y}\sim\mathcal{D}_\mathrm{T}|Q)} \left[ \tau(f_{\theta_\mathrm{S}}(\mathbf{x}_\delta), f_{\theta_\mathrm{T}}(\mathbf{y})) \right].$$

For example, if the adversary lacks knowledge of the precise caption in a text-to-image retrieval

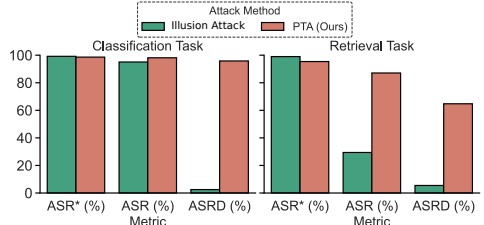

Figure 3: Comparison of attack performance in terms of **generalizability and undetectability**. ASR* represents the attack success rate when the true target is known (**ideal situation**), ASR corresponds to the attack success rate when the true target is unknown (measures **generalizability**), and ASRD denotes the attack success rate when the true target is unknown and anomaly detection is applied (further measures **undetectability**).

task, the adversarial image should be able to deceive semantically similar textual descriptions that match certain known keywords. For clarity, *generalizability* here refers to the ability of AEs to match

---

[2]More details of threat model based on retrieval and classification are provided in Appendix B.

partially known or semantically similar cross-modal targets. This differs from *transferability*, which measures the ability of AEs generated for one model to also fool another model (Gu et al., 2023).

In Figure 3, we illustrate the ASR of existing targeted attacks to match multiple semantically similar cross-modal targets. The results show that AEs crafted by the current method struggle to effectively align with multiple similar targets, which restricts their applicability in practical scenarios. This observation motivates us to explore strategies for improving the generalizability of AEs.

## 2.3 LIMITED UNDETECTABILITY OF THE TARGETED ATTACK

In what follows, we provide a general framework for detecting AEs in the embedding space and analyze adversarial undetectability using anomaly detection.

To well quantify the undetectability of AEs, we summarize a detection framework to identify the outliers likely to be adversarial. The detected outliers can be formalized as:

$$\mathbf{D}_{\text{outlier}} = \left\{ \mathbf{x}_i \mid s_i > \text{Quantile}(S, 1-r) \right\}_{i=1}^{N},$$

where $r$ is a pre-given anomaly ratio in unsupervised anomaly detection, $S = \{s_1, s_2, \ldots, s_N\}$ denotes a set of anomaly scores, and $\text{Quantile}(\cdot, 1-r)$ returns the value at the specified quantile of the anomaly scores. It is noteworthy that the calculation of the anomaly score $s$ varies depending on the detection method employed (Angiulli & Pizzuti, 2002; Breunig et al., 2000; Hoffmann, 2007; Liu et al., 2008), identifying those with higher scores as outliers.

Illusion Attack generates embeddings that lie far outside the benign data manifold (see Figure 2(a)), making them vulnerable to anomaly detectors. In Table 1, we show that existing targeted AEs can be effectively detected using simple anomaly detection methods such as $k$NN (Angiulli & Pizzuti, 2002), LOF (Breunig et al., 2000), Isolation Forest (Liu et al., 2008), and PCA (Hoffmann, 2007) in the embedding space, leading to a low Attack Success Rate (ASR) after anomaly detection (ASRD) as illustrated in Figure 3. This observation motivates us to explore methods to enhance the undetectability of AEs.

Table 1: **Average anomaly score ranks** ($\uparrow$ %) of AEs predicted by anomaly detectors (with 100 reference points used). The AEs were generated for LanguageBind (Zhu et al., 2024). **A larger rank indicates higher undetectability.**

| Task | Attack | $k$NN | LOF | Forest | PCA |
|---|---|---|---|---|---|
| Classification | Illusion Attack | 1.00 | 1.01 | 1.00 | 1.00 |
| | **PTA (Ours)** | **35.64** | **21.78** | **71.29** | **28.71** |
| Retrieval | Illusion Attack | 1.01 | 1.03 | 1.10 | 1.07 |
| | **PTA (Ours)** | **9.30** | **8.71** | **3.07** | **6.18** |

## 2.4 THE RELATIONSHIP BETWEEN THE GENERALIZABILITY AND UNDETECTABILITY

Here, we analyze how adversaries can generate AEs that achieve both high undetectability and generalizability in multimodal models. Since AEs lose their effectiveness once detected as anomalies, the adversary aims to maximize generalizability while remaining as undetectable as possible. If adversaries model the defender's anomaly detection algorithm in the embedding space as a distance-based outlier filtering problem (Angiulli & Pizzuti, 2002), the adversary's optimization objective for AE generation under anomaly detection defense can be formally defined as:

$$\min_{\mathbf{x}_\delta} \mathbb{E}_{\mathbf{y} \sim \mathcal{P}_{\text{target}}(\mathbf{Y} \sim \mathcal{D}_{\text{T}}|Q)} \left[ d\left( f_{\theta_{\text{S}}}(\mathbf{x}_\delta), f_{\theta_{\text{T}}}(\mathbf{y}) \right) \right]$$

$$\text{s.t.} \quad \mathbb{E}_{\mathbf{x} \sim \mathcal{P}_{\text{target}}(\mathbf{X} \sim \mathcal{D}_{\text{S}}|Q)} \left[ d\left( f_{\theta_{\text{S}}}(\mathbf{x}_\delta), f_{\theta_{\text{S}}}(\mathbf{x}) \right) \right] \leq \beta_{\text{true}},$$

where $d(\cdot)$ denotes a distance metric used in the anomaly detection algorithm, and $\beta_{\text{true}}$ is a threshold that distinguishes benign from anomalous examples. The objective aims to maximize the generalizability of $\mathbf{x}_\delta$ while ensuring it remains undetectable. However, in practice, adversaries do not know the exact value of $\beta_{\text{true}}$ set by the defender. Instead, they can only estimate $\beta$ as an approximation of the detection threshold. For analytical convenience, we employ the $L_2$ distance as the distance measure between the feature vectors, reformulating the optimization problem as:

$$\min_{\mathbf{x}_\delta} \mathbb{E}_{\mathbf{y}} \left[ \| f_{\theta_{\text{S}}}(\mathbf{x}_\delta) - f_{\theta_{\text{T}}}(\mathbf{y}) \|_2^2 \right]$$
$$\text{s.t.} \quad \mathbb{E}_{\mathbf{x}} \left[ \| f_{\theta_{\text{S}}}(\mathbf{x}_\delta) - f_{\theta_{\text{S}}}(\mathbf{x}) \|_2^2 \right] \leq \beta. \tag{1}$$

By solving Equation (1), the following relationship between generalizability and undetectability can be established, and the proof of Theorem 1 is provided in Appendix C:

**Theorem 1.** *Let* $\mathbf{v} = f_{\theta_S}(\mathbf{x}_\delta)$ *and define generalizability* $L(\mathbf{v}) = \mathbb{E}_\mathbf{y} \left[ \| \mathbf{v} - f_{\theta_T}(\mathbf{y}) \|_2^2 \right]$), *we have:*

$$\min_\mathbf{v} L(\mathbf{v}) = \left( \max \left\{ \| \mathbf{\Delta} \|_2 - \sqrt{\beta - \sigma_S}, 0 \right\} \right)^2 + \sigma_T,$$

*where* $\sigma_T = \mathrm{tr} \left( \mathrm{Var} \left[ f_{\theta_T}(\mathbf{y}) \right] \right)$ *and* $\sigma_S = \mathrm{tr} \left( \mathrm{Var} \left[ f_{\theta_S}(\mathbf{x}) \right] \right)$, *and* $\mathbf{\Delta} = \mathbb{E}_\mathbf{y}[f_{\theta_T}(\mathbf{y})] - \mathbb{E}_\mathbf{x}[f_{\theta_S}(\mathbf{x})]$ *represents the modality gap, as verified in Liang et al. (2022).*

Theorem 1 indicates that the optimal generalizability $L(\mathbf{v})$ is influenced by the modality gap $\| \mathbf{\Delta} \|_2$ and the estimated detection threshold $\beta$. Specifically, as $\| \mathbf{\Delta} \|_2$ decreases or $\beta$ increases, $L(\mathbf{v})$ reduces, enhancing the generalizability of the optimal AE. Furthermore, Theorem 1 reveals an interesting trade-off between generalizability and undetectability. Therefore, focusing only on target-modal data to improve generalizability inevitably compromises undetectability, as illustrated in Figure 2 (b). However, by incorporating source-modal targets to create multimodal proxies for optimizing the AEs, we are expected to derive AEs whose generalizability approaches the theoretical upper bound $L(\mathbf{v})$, while maintaining a fixed level of undetectability. This insight motivates us to design a new attack method that effectively balances these two factors, as discussed in the next section.

## 3 PTA: PROXY TARGETED ATTACK

In Sections 2.2 and 2.3, we identified the limitations of the existing targeted attack on undetectability and generalizability. Furthermore, Section 2.4 presents a theoretical analysis that reveals a limitation: these two challenges cannot be perfectly addressed simultaneously. To address these challenges, we propose Proxy Targeted Attack (PTA), endowing the AEs with both generalizability and undetectability. It introduces two key innovations: (i) leveraging multiple proxy targets to enhance generalizability, and (ii) optimizing AEs with respect to source-modal targets to improve undetectability.

To improve generalizability, we define the optimization loss $\mathcal{L}_G$ with $\mathbf{x}_\delta$ as:

$$\mathcal{L}_G(\mathbf{x}_\delta) = 1 - \frac{1}{N_c} \sum_{j=1}^{N_c} \tau \left( f_{\theta_S}(\mathbf{x}_\delta), f_{\theta_T}(\hat{\mathbf{y}}_j) \right), \tag{2}$$

where $\{ \hat{\mathbf{y}}_1, ..., \hat{\mathbf{y}}_{N_c} \}$ denotes a set of *target-modal proxies* that are sampled from the estimated ground-truth distribution $\mathcal{P}_{\text{target}}(\mathbf{Y} \sim \mathcal{D}_T | Q)$. Note that $N_c$ is a hyperparameter. We utilize this set of proxy targets to serve as surrogates for the unknown true target $\mathbf{y}_t$. By maximizing the cosine similarity, we can enhance the generalizability of AEs across multiple cross-modal targets, thereby increasing the likelihood of successful attacks on the true targets.

To improve undetectability, we define the undetectability optimization loss $\mathcal{L}_D$ for $\mathbf{x}_\delta$ as:

$$\mathcal{L}_D(\mathbf{x}_\delta) = \frac{1}{N_s} \sum_{i=1}^{N_s} \| f_{\theta_S}(\mathbf{x}_\delta) - f_{\theta_S}(\hat{\mathbf{x}}_i) \|_2, \tag{3}$$

where $\{ \hat{\mathbf{x}}_1, ..., \hat{\mathbf{x}}_{N_s} \}$ denotes a set of *source-modal proxies* that is sampled from the estimated distribution $\mathcal{P}_{\text{target}}(\mathbf{X} \sim \mathcal{D}_S | Q)$ and $N_s$ is the number of source-modal proxies. The objective is to position the AEs as close as possible to these source-modal proxies in the embedding space. Ideally, the AEs should lie within or near the convex polytope formed by benign examples, thereby enhancing concealment and minimizing the likelihood of detection.

Combining Equation (2) and Equation (3), the final optimization objective is defined as:

$$\arg\min_{\mathbf{x}_\delta} \mathcal{L}_G(\mathbf{x}_\delta) + \alpha \mathcal{L}_D(\mathbf{x}_\delta), \text{ s.t., } \| \mathbf{x}_\delta - \mathbf{x} \|_\infty \le \epsilon, \tag{4}$$

where $\mathbf{x}$ denotes the original example corresponding to $\mathbf{x}_\delta$ and the perturbation is constrained by a maximum perturbation limit $\epsilon$. The parameter $\alpha$ serves as a balancing factor between $\mathcal{L}_G$, and $\mathcal{L}_D$, allowing control over the dominance of the two abilities.

Furthermore, we demonstrate that using multiple proxies $\{ \hat{\mathbf{y}}_j \}_{j=1}^{N_c}$ and $\{ \hat{\mathbf{x}}_i \}_{i=1}^{N_s}$ can guarantee a lower bound on the generalizability performance of the AE, comparable with directly targeting the true target. We formalize this theoretical result by considering the effectiveness of source-modal proxies $\{ \hat{\mathbf{x}}_i \}_{i=1}^{N_s}$ in approximating the true target $\mathbf{y}_t$, in the following theorem:

**Theorem 2.** *Let* $\mathbf{x}_\delta$ *be the AE generated by using multiple source-modal proxies* $\{ \hat{\mathbf{x}}_i \}_{i=1}^{N_s}$ *and target-modal proxies* $\{ \hat{\mathbf{y}}_j \}_{j=1}^{N_c}$. *Let us denote by* $B_{N_s}$ *the empirical lower bound of the cosine similarity*

*between source-modal proxies and the true target, i.e., $B_{N_s} = \min_{i \in [N_s]} \tau\left(f_{\theta_S}(\hat{\mathbf{x}}_i), f_{\theta_T}(\mathbf{y}_t)\right)$. If $\mathbf{x}_\delta$ is an interior point of the convex polytope formed by the source-modal proxies $\{\hat{\mathbf{x}}_i\}_{i=1}^{N_s}$, then the similarity between the AE and the true target will be lower bounded by $B_{N_s}$, i.e.,*

$$\tau\left(f_{\theta_S}(\mathbf{x}_\delta), f_{\theta_T}(\mathbf{y}_t)\right) \geq B_{N_s}.$$

Theorem 2 provides theoretical support for the effectiveness of using multiple source-modal proxies to approximate the true target. It ensures that AEs maintain a high level of cosine similarity to the target distribution, thereby enhancing their generalization to unseen targets. Similarly, a corresponding theorem also holds for target-modal proxies. The formal proof and details are provided in Appendix D.

# 4 EXPERIMENTS

**Overview.** In this section, we first introduce our experimental setup. Then, we present a comparative analysis of our method against baseline approaches, focusing on the undetectability and generalizability of different adversarial attacks. Next, we discuss the effectiveness of PTA in more challenging scenarios (black-box attacks, textual or audio AEs, and potential defenses beyond anomaly detection). Finally, we analyze hyperparameter factors that could potentially affect our PTA.

## 4.1 EXPERIMENTAL SETTINGS

**Models, downstream tasks and datasets.** We use three recent multimodal pre-trained models: ImageBind (Girdhar et al., 2023), LanguageBind (Zhu et al., 2024), and One-PEACE (Wang et al., 2023a). ImageBind and LanguageBind support six modalities, while One-PEACE handles three modalities: image, text, and audio. As for downstream tasks, our experiments encompass two cross-modal matching tasks: classification and retrieval. For classification, we use ImageNet (Deng et al., 2009) and XmediaNet (Peng et al., 2018). For retrieval, we perform evaluations on MSCOCO (Lin et al., 2014) and XmediaNet. For these tasks, we generate AEs from the image modality, with true targets located in the text modality. We also evaluate situations where the AE are text or audio. Further details about models, tasks and datasets are provided in Appendix E.1.

**Compared baselines.** We compare PTA with prevailing targeted and untargeted attacks on multimodal pre-trained models. For *targeted attacks*, in addition to Illusion Attack (Zhang et al., 2024b), we also incorporate CrossFire (Dou et al., 2024) and MF-ii (Zhao et al., 2024b), which optimize AEs with the source-modal example generated based on the cross-modal target. For *untargeted attacks*, we compare PTA with Sep-Attack (Madry et al., 2019; Li et al., 2020a), Co-Attack (Zhang et al., 2022), SGA (Lu et al., 2023), and CMI-Attack (Fu et al., 2024) from the perspective of poisoned retrieval system performance degradation. Details of these baselines are in Appendix E.2.

**Metrics.** For classification, we assess the adversarial attacks using the Classification Attack Success Rate (*Cls ASR*), which quantifies the percentage of AEs successfully classified as the target class. In retrieval, we evaluate performance using the Recall at Rank K Attack Success Rate (*R@K ASR*), measuring the proportion of AEs retrieved within the top-K results that match the true target. With anomaly detection methods: Angiulli & Pizzuti (2002); Breunig et al. (2000); Liu et al. (2008); Hoffmann (2007) enabled, we report *Cls ASRD* and *R@K ASRD*, i.e., the corresponding ASR computed over *undetected* AEs. Specifically, we evaluate the effectiveness of PTA along two axes:

- To evaluate the generalizability, we test the performance (*Cls ASR* or *R@K ASR*) of AEs in zero-shot classification (text as prompt) and text-to-image retrieval tasks.

- To measure the undetectability, we apply traditional anomaly detection methods along with our proposed anomaly detection approach in Section 2.3. We assess the performance (*Cls ASRD* or *R@K ASRD*) of AEs that bypass detection.

More details of the evaluation metrics are provided in Appendix E.3.

**Hyperparameters.** For all the anomaly detection methods, we set $K = 50$ to denote the number of top-$K$ samples, with the filtering ratio $r = 1 - \frac{N_{\text{adv}}}{K}$, where $N_{\text{adv}}$ is the number of AEs within the selected top-$K$ range for evaluation convenience. All results are reported as the average performance across three runs with different random seeds. We use the PGD attack (Madry et al., 2019) under the $L_\infty$-norm with 100 iterations and $\epsilon = 8/255$ for both classification and retrieval tasks; More details on the hyperparameters are provided in Appendix E.4. Specifically, experiments in Sections 4.2 and 4.3 both follow these settings unless otherwise stated.

Table 2: Comparison of **generalizability and undetectability** of AEs in classification task. Performance is reported by *Cls ASR (%)* and *Cls ASRD (%)* when anomaly-detection defense is used.

| Attack | Defense | XmediaNet | | | ImageNet | | | Average |
|---|---|---|---|---|---|---|---|---|
| | | ImageBind | LanguageBind | One-PEACE | ImageBind | LanguageBind | One-PEACE | |
| MF-ii | ✗ | $30.89_{0.63}$ | $48.00_{4.71}$ | $36.67_{3.31}$ | $17.33_{15.04}$ | $57.78_{10.75}$ | $32.90_{0.61}$ | 37.26 |
| CrossFire | ✗ | $31.33_{0.54}$ | $45.56_{6.49}$ | $38.00_{3.93}$ | $12.89_{6.07}$ | $53.33_{3.93}$ | $29.60_{0.49}$ | 35.12 |
| Illusion Attack | ✗ | $\mathbf{99.58}_{0.59}$ | $\mathbf{100.00}_{0.00}$ | $97.50_{0.00}$ | $77.44_{0.21}$ | $95.06_{0.69}$ | $89.50_{0.64}$ | 93.18 |
| **PTA (Ours)** | ✗ | $\mathbf{99.58}_{0.59}$ | $\mathbf{100.00}_{0.00}$ | $\mathbf{98.75}_{0.59}$ | $\mathbf{94.22}_{0.47}$ | $\mathbf{99.72}_{0.20}$ | $\mathbf{97.61}_{0.78}$ | **98.31** |
| MF-ii | ✓ | $20.89_{7.78}$ | $43.69_{2.70}$ | $36.67_{3.31}$ | $6.67_{7.62}$ | $35.33_{2.37}$ | $28.70_{0.52}$ | 28.66 |
| CrossFire | ✓ | $31.33_{0.54}$ | $42.16_{4.79}$ | $35.97_{2.74}$ | $9.33_{8.56}$ | $41.78_{9.11}$ | $25.10_{0.40}$ | 30.95 |
| Illusion Attack | ✓ | $18.33_{0.00}$ | $18.33_{1.18}$ | $66.25_{0.59}$ | $0.42_{0.59}$ | $2.50_{0.00}$ | $14.42_{0.15}$ | 20.04 |
| **PTA (Ours)** | ✓ | $\mathbf{92.92}_{0.59}$ | $\mathbf{95.42}_{0.59}$ | $\mathbf{87.92}_{0.59}$ | $\mathbf{77.54}_{1.31}$ | $\mathbf{95.82}_{0.73}$ | $\mathbf{55.42}_{1.33}$ | **84.17** |

Table 3: Comparison of **generalizability and undetectability** of AEs in retrieval task. Performance is reported by *R@1 ASR (%)* and *R@1 ASRD (%)* when anomaly-detection defense is used.

| Attack | Defense | XmediaNet | | | MSCOCO | | | Average |
|---|---|---|---|---|---|---|---|---|
| | | ImageBind | LanguageBind | One-PEACE | ImageBind | LanguageBind | One-PEACE | |
| MF-ii | ✗ | $0.00_{0.00}$ | $0.00_{0.00}$ | $0.00_{0.00}$ | $0.00_{0.00}$ | $0.00_{0.00}$ | $0.00_{0.00}$ | 0.00 |
| CrossFire | ✗ | $0.00_{0.00}$ | $0.00_{0.00}$ | $0.00_{0.00}$ | $0.00_{0.00}$ | $0.00_{0.00}$ | $0.00_{0.00}$ | 0.00 |
| Illusion Attack | ✗ | $77.05_{0.47}$ | $85.80_{1.72}$ | $56.76_{1.33}$ | $20.33_{1.30}$ | $29.41_{0.75}$ | $9.59_{0.18}$ | 46.49 |
| **PTA (Ours)** | ✗ | $\mathbf{95.36}_{0.27}$ | $\mathbf{96.75}_{0.04}$ | $\mathbf{85.14}_{0.12}$ | $\mathbf{71.31}_{0.71}$ | $\mathbf{87.09}_{0.13}$ | $\mathbf{30.69}_{0.18}$ | **77.72** |
| MF-ii | ✓ | $0.00_{0.00}$ | $0.00_{0.00}$ | $0.00_{0.00}$ | $0.00_{0.00}$ | $0.00_{0.00}$ | $0.00_{0.00}$ | 0.00 |
| CrossFire | ✓ | $0.00_{0.00}$ | $0.00_{0.00}$ | $0.00_{0.00}$ | $0.00_{0.00}$ | $0.00_{0.00}$ | $0.00_{0.00}$ | 0.00 |
| Illusion Attack | ✓ | $15.98_{0.33}$ | $2.53_{0.80}$ | $45.06_{0.18}$ | $10.39_{0.11}$ | $5.47_{1.72}$ | $9.59_{0.18}$ | 14.84 |
| **PTA (Ours)** | ✓ | $\mathbf{74.94}_{2.67}$ | $\mathbf{76.64}_{0.20}$ | $\mathbf{76.22}_{0.24}$ | $\mathbf{50.11}_{0.64}$ | $\mathbf{64.75}_{0.35}$ | $\mathbf{28.13}_{0.09}$ | **61.80** |

Table 4: Comparison of **text-to-image retrieval degradation** by injecting varying fractions of AEs. *Injection Ratio* is the proportion of AEs to all images. Results are reported as *R@1 (%) after injection (↓ drop in R@1 (%))*. Here, *R@1 (Recall@1)* is the fraction of queries whose top-ranked result is its corresponding ground-truth image. **Lower R@1 indicates stronger disruption brought by AEs.**

| Attack (Injection ratio) | ImageBind | LanguageBind | One-PEACE |
|---|---|---|---|
| No Attack (0) | *41.02* | *39.62* | *37.47* |
| Sep-Attack (10%) | 38.54 (↓ 2.48) | 37.38 (↓ 2.24) | 35.73 (↓ 1.74) |
| Co-Attack (10%) | 37.34 (↓ 3.68) | 35.69 (↓ 3.93) | 34.31 (↓ 3.16) |
| SGA (10%) | 36.86 (↓ 4.16) | 35.54 (↓ 4.08) | 33.91 (↓ 3.56) |
| CMI-Attack (10%) | 36.90 (↓ 4.12) | 36.00 (↓ 3.62) | 33.80 (↓ 3.67) |
| Illusion Attack (1%) | 37.42 (↓ 3.60) | 34.55 (↓ 5.07) | 36.55 (↓ 0.92) |
| **Our PTA** (0.1%) | 33.27 (↓ 7.75) | 36.23 (↓ 3.39) | 37.26 (↓ 0.21) |
| **Our PTA** (0.5%) | 23.29 (↓ 17.73) | 15.47 (↓ 24.15) | 33.55 (↓ 3.09) |
| **Our PTA** (1%) | **20.04** (↓ **20.98**) | **12.26** (↓ **27.36**) | **32.83** (↓ **4.64**) |

## 4.2 THE EFFECTIVENESS OF PTA

In this part, we evaluate the generalizability and undetectability of AEs generated by PTA in a white-box setting. To evaluate the *generalizability*, we optimize AEs knowing the targeted estimated distribution $\mathcal{P}_{\text{target}}(\mathbf{Y} \sim \mathcal{D}_{\text{T}}|Q)$ but without complete details about the true targets. We select two disjoint subsets from the distribution to serve as proxy targets and true targets, allowing us to assess the generalizability performance of the AEs. Also, we evaluate the *undetectability* of AEs by applying our anomaly detection tailored for multimodal embeddings. Detailed settings are in Section 4.1.

**Classification task.** As shown in Table 2, PTA surpasses the adopted baselines by a large margin. We also observe that multimodal classification systems are more vulnerable than retrieval systems (shown in Table 3). We conjecture that it is because the target distribution in classification (i.e., class prompts) is less sparse than in retrieval (i.e., user queries), making alignment feasible even for less generalizable AEs. This suggests that increasing the variety and diversity of class prompts could potentially improve adversarial robustness against generalized AEs. More discussion of this vulnerability and possible explanation is provided in Appendix F.1.

**Retrieval task.** As shown in Table 3, our approach substantially improves both ASR and ASR under anomaly detection (ASRD) over multiple baselines (Zhang et al., 2024b; Dou et al., 2024; Zhao et al., 2024b). This is because using multiple cross-modal proxy targets enhances AE generalizability to semantically similar texts, while incorporating source-proxy targets tightens alignment with the source modality and improves embedding stealthiness. Further, we quantify the risk of highly generalizable AEs in retrieval systems by considering injecting AEs into the image gallery as poison to

Table 5: Comparison results of **generalizability** of AEs in **black-box attacks**. Results are reported for different tasks (*Cls ASR (%)* for classification and *R@1 ASR (%)* for retrieval).

| | | Classification | | | | Retrieval | | | |
| | | XmediaNet | | ImageNet | | XmediaNet | | MSCOCO | |
| Attack | Queries | ImageBind | LanguageBind | ImageBind | LanguageBind | ImageBind | LanguageBind | ImageBind | LanguageBind |
|---|---|---|---|---|---|---|---|---|---|
| Illusion Attack | $10^4$ | $49.58_{1.77}$ | $65.00_{0.00}$ | $33.11_{1.27}$ | $49.51_{0.93}$ | $36.96_{0.74}$ | $39.57_{1.72}$ | $7.86_{0.11}$ | $8.22_{0.53}$ |
| **PTA (Ours)** | $10^4$ | $\mathbf{51.67}_{1.18}$ | $\mathbf{68.75}_{2.95}$ | $\mathbf{34.75}_{0.89}$ | $\mathbf{52.44}_{0.88}$ | $\mathbf{60.72}_{5.10}$ | $\mathbf{61.74}_{1.39}$ | $\mathbf{20.77}_{1.43}$ | $\mathbf{27.63}_{2.80}$ |
| Illusion Attack | $2 \cdot 10^4$ | $75.00_{0.00}$ | $87.08_{0.59}$ | $59.87_{1.35}$ | $75.96_{0.02}$ | $61.35_{0.00}$ | $60.93_{0.00}$ | $11.56_{0.00}$ | $11.64_{0.64}$ |
| **PTA (Ours)** | $2 \cdot 10^4$ | $\mathbf{78.33}_{0.00}$ | $\mathbf{91.67}_{0.00}$ | $\mathbf{64.32}_{3.54}$ | $\mathbf{81.75}_{1.45}$ | $\mathbf{84.72}_{0.00}$ | $\mathbf{83.00}_{0.00}$ | $\mathbf{41.81}_{0.00}$ | $\mathbf{50.14}_{1.22}$ |

Table 6: Comparison results of **generalizability and undetectability** of **textual or audio** AEs. We report the performance of AEs by *Cls ASR (%)* (*Cls ASRD (%)* with defense) and *R@1 ASR (%)* (*Cls ASRD (%)* with defense) for classification and retrieval.

| Task | Method | Source Modality: Text | | Source Modality: Audio | |
| | | No Defense | With Defense | No Defense | With Defense |
|---|---|---|---|---|---|
| Classification | Illusion Attack | $26.01_{0.35}$ | $10.54_{0.21}$ | $\mathbf{100.00}_{0.00}$ | $21.45_{0.55}$ |
| | **PTA (Ours)** | $\mathbf{37.32}_{0.29}$ | $\mathbf{25.88}_{0.40}$ | $\mathbf{100.00}_{0.00}$ | $\mathbf{91.03}_{0.78}$ |
| Retrieval | Illusion Attack | $10.67_{0.03}$ | $1.35_{0.15}$ | $0.26_{0.04}$ | $0.05_{0.01}$ |
| | **PTA (Ours)** | $\mathbf{24.33}_{0.27}$ | $\mathbf{18.91}_{0.32}$ | $\mathbf{65.33}_{0.17}$ | $\mathbf{48.17}_{0.25}$ |

compromise the *overall* retrieval performance on MSCOCO (Lin et al., 2014). Specifically, different from Table 3, we test the system with *all* queries in examples of MSCOCO. Unlike untargeted AEs, which typically break only the link between their single query, generalized targeted AEs attract many more semantically related queries. As a result, PTA causes markedly larger performance degradation with fewer injected AEs (Table 4) than four recent untargeted attacks (Madry et al., 2019; Li et al., 2020a; Zhang et al., 2022; Lu et al., 2023; Fu et al., 2024) and Illusion Attack (Zhang et al., 2024b), demonstrating its high attack effectiveness and efficiency.

### 4.3 THE EFFECTIVENESS OF PTA UNDER MORE CHALLENGING CONDITIONS

Here, we assess PTA's performance in tougher conditions: (i) in black-box settings, (ii) with textual or audio AEs, and (iii) against defenses of adversarial training, data augmentation, or adversarial purification. *Since source-modal target optimization methods (Dou et al., 2024; Zhao et al., 2024b) perform poorly even in the easiest settings, they will not be considered in this part.*

**Black-box attacks.** In scenarios where the adversaries have query access to the encoder but no direct access to the model weights, AEs can still be generated using a limited number of queries. This bypasses the need for gradient information, making it possible to conduct a black-box attack through estimated gradients or random search techniques. In our black-box setting, we experiment with gradient-free Square Attack (Andriushchenko et al., 2020) under the $L_\infty$-norm with $\epsilon = 16/255$. We evaluate our approach against Illusion Attack under an equal query budget, with results in Table 5. Across classification and retrieval tasks, our method also achieves superior performance, highlighting its superior generalizability even in black-box scenarios.

**Textual or audio adversarial examples.** For textual AEs, we use Bert-Attack (Li et al., 2020b) with a perturbation budget of 10% of tokens and evaluate image-text retrieval and text classification on MSCOCO (for classification, images serve as labels). For audio AEs, we apply PGD with an $\ell_\infty$ budget of 0.01 and evaluate audio-text retrieval and audio classification on XmediaNet (for classification, text prompts serve as labels). Results in Table 6 show that discrete text AEs are indeed harder to optimize than image AEs, yielding lower ASR. Nevertheless, PTA consistently improves both generalizability and undetectability. In addition, continuous audio AEs are as effective as image AEs, and PTA again brings substantial gains in both generalizability and undetectability.

**Possible defenses.** Here, we also evaluate the effectiveness of PTA under possible defense methods in addition to anomaly detection. Specifically, we adopt three prevailing defenses against adversarial attacks in retrieval tasks: (i) *TeCoA* (Mao et al., 2023), a state-of-the-art method for adversarial training on pretrained vision-language models. (ii) *Data aug-*

Table 7: *R@1 ASR (%)* under three defenses: Adversarial training (AT), data augmentation (DA) and adversarial purification (AP).

| Method | Defense | | |
| | AT | DA | AP |
|---|---|---|---|
| Illusion Attack | $62.31_{0.24}$ | $12.44_{0.13}$ | $9.83_{0.07}$ |
| **PTA (Ours)** | $\mathbf{78.03}_{0.20}$ | $\mathbf{89.33}_{0.31}$ | $\mathbf{71.97}_{0.37}$ |

*mentation* that augments input to disrupt adversarial features. We use Gaussian Blur here. (iii) *DiffPure* (Nie et al., 2022) that adopts diffusion models (Ho et al., 2020) for adversarial input purification. The results are shown in Table 7, which reflects the effectiveness of PTA against not only anomaly detection, but also other defenses. We hypothesize that this stems from PTA's generalizability: by aligning to a distribution of semantically consistent targets via proxies, PTA maintains high ASR even when brittle adversarial features are attenuated by defense. Due to space constraints, we defer the experiment configurations and extended results in Appendix E.5.

We also study the effectiveness of PTA with *limited adversarial prior knowledge* (Appendix F.2), *audio target modality* (Appendix F.3), and *unknown target modality* (Appendix F.4) in the Appendix.

### 4.4 ABLATION STUDIES

**Number of proxy targets.** Figure 4 shows that increasing the number of target-modal or source-modal proxies can improve the attack effectiveness of AEs. In specific, increasing target-modal proxies markedly boosts ASR in retrieval but only modestly in classification, likely due to the lower similarity between target-modal proxies in retrieval tasks, making the training samples more versatile. In summary, a few dozen proxies suffice for strong performance in both retrieval and classification. We also analyze the cost of gathering and optimizing with proxies in Appendix F.5.

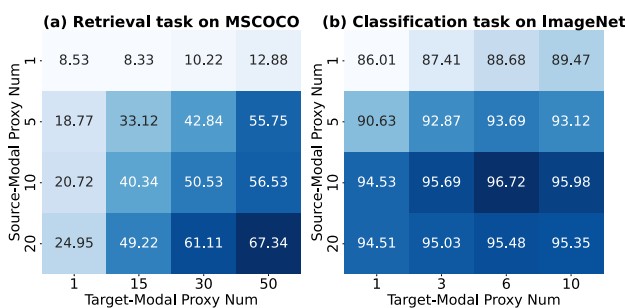

Figure 4: Attack performance with different number of **target-modal proxies** ($N_c$) and **source-modal proxies** ($N_s$).

**Balancing factor $\alpha$.** In Equation (4), $\alpha$ is critical for controlling the alignment of AEs with the target modality versus the source modality. The ASR-ASRD trade-off observed in Figure 5 aligns with our theoretical analysis in Section 2.4 and gives practitioners a way to precisely tune their objective: lower $\alpha$ emphasizes broad cross-target matching, whereas higher $\alpha$ emphasizes stealth. The effect is more pronounced for retrieval, where targets are more dispersed (empirically demonstrated in Appendix F.1), but the same tuning rule holds across tasks. To summarize, practitioners can set a lower $\alpha$ to prioritize generalizability and a higher $\alpha$ to prioritize undetectability.

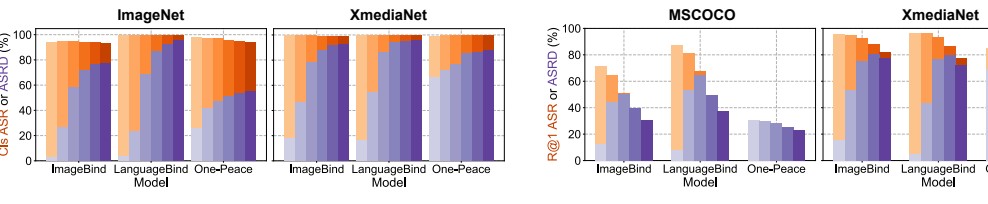

Classification, $\alpha \in \{0, 0.2, 0.4, 0.6, 0.8, 1.0\}$      Retrieval, $\alpha \in \{0, 0.2, 0.4, 0.6, 0.8\}$

Figure 5: Analysis of the **balancing factor** $\alpha$. We present results for two metrics: ASR (Attack Success Rate (%)) and ASRD (Attack Success Rate after anomaly Detection (%)). ASR (red) quantifies the generalizability of AEs, while ASRD (purple) measures their undetectability.

## 5 CONCLUSION

In this paper, we investigated targeted adversarial attacks in cross-modal matching tasks by examining both undetectability and generalizability. Our anomaly detection analysis in the embedding space reveals that existing targeted AEs are vulnerable to detection and exhibit poor generalization to semantically similar or partially known targets. To address these challenges, we proposed Proxy Targeted Attack (PTA), which leverages multimodal proxies to achieve both superior undetectability and generalizability. In addition, our theoretical findings highlight the interplay between these two limitations of AEs and demonstrate how PTA achieves an optimal balance between them. Experiments validate PTA's effectiveness in generating undetectable AEs while maintaining a high success rate against semantically similar targets, underscoring its potential for real-world adversarial scenarios.

ETHICS STATEMENT

This work examines targeted adversarial attacks on multimodal pre-trained models to better understand and mitigate security risks in multimodal systems. We highlight the risks observed in this paper (e.g., generalizable and hard-to-detect AEs) to alert practitioners that such attacks are feasible and to motivate stronger defenses. No production systems or personal data are involved.

REPRODUCIBILITY STATEMENT

An anonymized repository accompanies this paper with code to reproduce results. Experiments rely on public datasets (ImageNet, MSCOCO, XmediaNet) with standard splits and official checkpoints of ImageBind, LanguageBind, and One-PEACE. Default hyperparameters, precise metric definitions, and step-by-step evaluation procedures are provided in the paper and the repository.

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

# Appendix for *Improving Generalizability and Undetectability for Targeted Adversarial Attacks on Multimodal Pre-trained Models*

We summarize the Appendix as follows:

# A    RELATED WORK

## A.1    MULTIMODAL PRE-TRAINED MODELS

Multimodal pre-trained models have garnered increasing interest for their ability to integrate diverse input modalities, such as images, text, and audio, into a unified latent space. These models serve as foundational representation encoders, enabling various downstream applications (Girdhar et al., 2023; Zhu et al., 2024; Wang et al., 2023a; Guzhov et al., 2022), or as multimodal processing modules integrated into high-level models (Su et al., 2023; Xing et al., 2024). Typically, these models are trained using contrastive learning (van den Oord et al., 2019) on multimodal paired datasets, such as image-text or audio-text pairs (Schuhmann et al., 2021; Kim et al., 2019). By maximizing the similarity between positive pairs while minimizing it for negative pairs, these models learn effective representations in the embedding space, where semantically similar inputs are mapped closer together. To enhance flexibility and model capacity, existing multimodal pre-trained models often employ dedicated encoders for each modality. For instance, ImageBind (Girdhar et al., 2023) and LanguageBind (Zhu et al., 2024) use separate transformers (Vaswani et al., 2017) for their six supported input modalities. Alternatively, models like One-PEACE (Wang et al., 2023a) adopt a hybrid approach, incorporating both modality-specific parameters and shared cross-modal parameters to process multimodal inputs. In this work, we evaluate both types of models in our experiments.

## A.2    ADVERSARIAL ATTACKS ON MULTIMODAL MODELS

Adversarial attacks and related security challenges on multimodal models have drawn significant attention (Tu et al., 2023; Vatsa et al., 2023). Compared with traditional single-modal models, the complexity and diversity of multimodal models make them more susceptible to adversarial attacks (Fan et al., 2024; Liu et al., 2024b; Zhao et al., 2024b). Prior research has predominantly focused on **untargeted adversarial attacks** against multimodal models (Zhang et al., 2022; Zhou et al., 2023; Lu et al., 2023; Wang et al., 2024a; 2023b), particularly Vision-Language Models (VLMs) like CLIP (Radford et al., 2021). These attacks typically perturb both text and image inputs to force the model into incorrect predictions or undesirable output. In this work, we focus on **targeted adversarial attacks**, where the adversary has a specific goal and aims to steer the model's output toward a designated target. Targeted attacks are more challenging than untargeted attacks (Zhao et al., 2024c; Li et al., 2020c), as they require precise alignment across multiple modalities to generate highly adversarial examples. Pioneering work (Zhang et al., 2024b) first explored the generation of targeted AEs for multimodal models, demonstrating their feasibility and effectiveness.

## A.3    DEFENSE MECHANISMS FOR MULTIMODAL MODELS

The multimodal models have highlighted their sensitivity to adversarial attacks, driving the development of defense mechanisms tailored to them (Zhao et al., 2024a; Liu et al., 2024a). Unlike single-modal models, multimodal systems must account for the interactions between different modalities, necessitating specialized defense strategies. Previous studies have primarily focused on adversarial fine-tuning for multimodal pre-trained models. These approaches include partial fine-tuning, such as text prompt tuning (Li et al., 2024; Zhang et al., 2024a) and visual prompt tuning (Mao et al., 2023), as well as full-parameter fine-tuning of the models (Wang et al., 2024b; Mao et al., 2023; Schlarmann et al., 2024; Waseda & Tejero-de-Pablos, 2024; Zhou et al., 2024). For instance, the pioneering work TeCoA (Mao et al., 2023) employs a text-guided contrastive adversarial training loss to fine-tune pre-trained multimodal models, enhancing their zero-shot adversarial robustness. However, a persistent challenge with adversarial fine-tuning is the trade-off between robustness and performance on benign examples (Mao et al., 2023; Zhang et al., 2019; Raghunathan et al., 2019). Our work adopts a novel perspective by emphasizing the detection of AEs as a defense strategy, mitigating their influence on the model's outputs.

# B    EXPLANATIONS OF THE THREAT MODEL

## B.1    THREAT MODEL IN CLASSIFICATION

In classification tasks, the adversary injects adversarial perturbations into a user input to steer the prediction toward a target class. Unlike conventional supervised classifiers, where each target is

a fixed label, multimodal classifiers induce class embeddings via class prompts **dynamically** and **privately** specified by downstream practitioners. For example, the class *elephant* may be encoded by prompts such as "A photo of an elephant" or "A huge elephant", which are inaccessible to the adversary. Consequently, we assume the adversary knows only the coarse class concept (e.g., *elephant*) rather than the exact prompt design, reflecting a limited prior-knowledge scenario. Under this assumption, the goal is to craft AEs that generalize across a distribution of plausible prompt formulations rather than overfit to a single, known prompt.

A practical example is **content moderation systems** that raise alarms for sensitive categories (e.g., weapons, explicit content) and remain silent for non-sensitive classes (e.g., animals, people). The textual prompts defining these categories (e.g., "a person," "a photo of an elephant," "a pistol") are typically private to enhance coverage and robustness. An attacker cannot access the exact prompts; nevertheless, an adversarially perturbed handgun image that generalizes across unseen prompts may be *misclassified* into a benign class (e.g., "a person" or "an elephant"), suppressing the alarm and exposing users to prohibited content.

### B.2 THREAT MODEL IN RETRIEVAL

In retrieval tasks, the adversary seeks to cause queries for a target concept to retrieve attacker-controlled AEs injected into the gallery of a multimodal retrieval system. Crucially, AEs must be crafted **prior to** the user's **dynamic** query target and thus it is impossible for the adversary to know the exact phrasing of the user query. Accordingly, as in classification, the adversary is assumed to know only the coarse concept of plausible queries (e.g., *elephant*), not their precise prompt formulations—again reflecting limited prior knowledge.

A practical instance is **multimodal search** (e.g., systems built with Amazon OpenSearch Service and Titan Multimodal Embeddings), where sellers upload product images to an open catalog and users issue text queries. The system embeds text and images into a shared vector space for text-to-image retrieval. An adversary as a seller can upload adversarially crafted images and probe embeddings via APIs; by optimizing these images (e.g., with PTA) toward a broad concept (e.g., *handbag*), legitimate queries related to that concept are more likely to retrieve the attacker's pre-crafted AEs. This manipulation diverts attention from authentic items and can lead to exposure of counterfeit goods, unfair competition, and misinformation.

## C PROOF OF THE RELATIONSHIP BETWEEN UNDETECTABILITY AND GENERALIZABILITY

For a random vector $\mathbf{y}$ and a given vector $\mathbf{x}$, we have the following lemma:
**Lemma C.1.**
$$\mathbb{E}_{\mathbf{y}}\left[\|\mathbf{x} - \mathbf{y}\|_2^2\right] = \|\mathbf{x} - \mathbb{E}_{\mathbf{y}}[\mathbf{y}]\|_2^2 + \mathrm{Var}[\mathbf{y}].$$
**Proof 1.**
$$\mathbb{E}_{\mathbf{y}}\left[\|\mathbf{x} - \mathbf{y}\|_2^2\right]$$
$$= \mathbb{E}_{\mathbf{y}}\left[\|\mathbf{x} - \mathbb{E}_{\mathbf{y}}[\mathbf{y}] + \mathbb{E}_{\mathbf{y}}[\mathbf{y}] - \mathbf{y}\|_2^2\right]$$
$$= \mathbb{E}_{\mathbf{y}}\left[\|\mathbf{x} - \mathbb{E}_{\mathbf{y}}[\mathbf{y}]\|_2^2 + 2(\mathbf{x} - \mathbb{E}_{\mathbf{y}}[\mathbf{y}])^\top (\mathbb{E}_{\mathbf{y}}[\mathbf{y}] - \mathbf{y})\right.$$
$$\left. + \|\mathbb{E}_{\mathbf{y}}[\mathbf{y}] - \mathbf{y}\|_2^2\right]$$
$$= \|\mathbf{x} - \mathbb{E}_{\mathbf{y}}[\mathbf{y}]\|_2^2 + 2(\mathbf{x} - \mathbb{E}_{\mathbf{y}}[\mathbf{y}])^\top \mathbb{E}_{\mathbf{y}}[\mathbb{E}_{\mathbf{y}}[\mathbf{y}] - \mathbf{y}]$$
$$+ \mathbb{E}_{\mathbf{y}}\left[\|\mathbb{E}_{\mathbf{y}}[\mathbf{y}] - \mathbf{y}\|_2^2\right]$$
$$= \|\mathbf{x} - \mathbb{E}_{\mathbf{y}}[\mathbf{y}]\|_2^2 + \mathrm{tr}\left(\mathrm{Var}[\mathbf{y}]\right)$$
$$\square$$

For the purpose of facilitating the derivation process, we denote $\mathbb{E}_{\mathbf{x}}$ as $\mathbb{E}_{\mathbf{x} \sim \mathcal{P}_{\text{target}}(\mathbf{X} \sim \mathcal{D}_{\text{S}}|Q)}$ and $\mathbb{E}_{\mathbf{y}}$ as $\mathbb{E}_{\mathbf{y} \sim \mathcal{P}_{\text{target}}(\mathbf{Y} \sim \mathcal{D}_{\text{T}}|Q)}$. Then, we can reformulate the optimization objective as follows:
$$\min_{x_\delta} \mathbb{E}_{\mathbf{y}}\left[\|f_{\theta_{\text{S}}}(\mathbf{x}_\delta) - f_{\theta_{\text{T}}}(\mathbf{y})\|_2^2\right]$$
$$\text{s.t.} \quad \mathbb{E}_{\mathbf{x}}\left[\|f_{\theta_{\text{S}}}(\mathbf{x}_\delta) - f_{\theta_{\text{S}}}(\mathbf{x})\|_2^2\right] \le \beta.$$

According to Lemma 1, we have:

$$\min_{\mathbf{v}} L(\mathbf{v}) = \|\mathbf{v} - \mu_\mathrm{T}\|_2^2 + \sigma_\mathrm{T}$$

$$\text{s.t.} \quad \|\mathbf{v} - \mu_\mathrm{S}\|_2^2 + \sigma_\mathrm{S} \leq \beta.$$

where $\mathbf{v} = f_{\theta_\mathrm{S}}(\mathbf{x}_\delta)$, $\mu_\mathrm{T} = \mathbb{E}_\mathbf{y}\left[f_{\theta_\mathrm{T}}(\mathbf{y})\right]$, $\mu_\mathrm{S} = \mathbb{E}_\mathbf{x}\left[f_{\theta_\mathrm{S}}(\mathbf{x})\right]$, $\sigma_\mathrm{T} = \mathrm{tr}\left(\mathrm{Var}\left[f_{\theta_\mathrm{T}}(\mathbf{y})\right]\right)$ and $\sigma_\mathrm{S} = \mathrm{tr}\left(\mathrm{Var}\left[f_{\theta_\mathrm{S}}(\mathbf{x})\right]\right)$. By applying the Lagrange multiplier method, we construct the Lagrangian function as follows:

$$F(\mathbf{v}^\star, \lambda, m) = \|\mathbf{v} - \mu_\mathrm{T}\|_2^2 + \sigma_\mathrm{T}$$
$$+ \lambda\left(\|\mathbf{v} - \mu_\mathrm{S}\|_2^2 + \sigma_\mathrm{S} + m^2 - \beta\right).$$

By taking the derivative with respect to each variable and setting the result equal to zero, we obtain the solution as follows:

$$\mathbf{v} = \begin{cases} \frac{\beta \boldsymbol{\Delta}}{\|\boldsymbol{\Delta}\|_2} + \mu_S & \text{if } \|\boldsymbol{\Delta}\|_2 > \sqrt{\beta - \sigma_\mathrm{S}} \\ \mu_T & \text{if } \|\boldsymbol{\Delta}\|_2 \leq \sqrt{\beta - \sigma_\mathrm{S}}. \end{cases}$$

where $\|\boldsymbol{\Delta}\|_2 = \|\mu_\mathrm{T} - \mu_\mathrm{S}\|_2$ denotes the modality gap. Therefore, the minimum value of $L(\mathbf{v})$ is:

$$L(\mathbf{v}^\star) = \left(\max\left\{\|\boldsymbol{\Delta}\|_2 - \sqrt{\beta - \sigma_\mathrm{S}}, 0\right\}\right)^2 + \sigma_\mathrm{T},$$

# D    PROOF OF EFFECTIVENESS FOR PROXY TARGETS

## D.1    THEOREM 2 FOR SOURCE-MODAL PROXIES

Since $\mathbf{x}_\delta$ is an interior point of the convex polytope formed by the source-modal proxies, we can express $\mathbf{x}_\delta$ as a convex combination of the proxy targets:

$$\mathbf{x}_\delta = \sum_{i=1}^{N_s} \beta_i \hat{\mathbf{x}}_i, \tag{5}$$

where $\beta_i \geq 0$ for all $i$, and $\sum_{i=1}^{N_s} \beta_i = 1$.

For the dot product $\mathbf{x}_\delta \cdot \mathbf{y}_t$, expressing $\mathbf{x}_\delta$ with Equation (5), we have:

$$\mathbf{x}_\delta \cdot \mathbf{y}_t = \left(\sum_{i=1}^{N_s} \beta_i \hat{\mathbf{x}}_i\right) \cdot \mathbf{y}_t = \sum_{i=1}^{N_s} \beta_i \left(\hat{\mathbf{x}}_i \cdot \mathbf{y}_t\right). \tag{6}$$

Given that $\tau\left(\hat{\mathbf{x}}_i, \mathbf{y}_t\right) \geq B_{N_s}$ for all $i \in [N_s]$ and $\tau$ representing the cosine similarity, it follows that:

$$\hat{\mathbf{x}}_i \cdot \mathbf{y}_t \geq B_{N_s} \|\hat{\mathbf{x}}_i\| \|\mathbf{y}_t\| \quad \text{for all } i \in [N_s]. \tag{7}$$

Substituting $\hat{\mathbf{x}}_i \cdot \mathbf{y}_t$ in Equation (6) with Equation (7), we obtain inequality:

$$\mathbf{x}_\delta \cdot \mathbf{y}_t \geq \sum_{i=1}^{N_s} B_{N_s} \beta_i \|\hat{\mathbf{x}}_i\| \|\mathbf{y}_t\|. \tag{8}$$

The cosine similarity between $\mathbf{x}_\delta$ and $\mathbf{y}_t$ can be represented with Equation (8) as:

$$\tau\left(\mathbf{x}_\delta, \mathbf{y}_t\right) = \frac{\mathbf{x}_\delta \cdot \mathbf{y}_t}{\|\mathbf{x}_\delta\| \|\mathbf{y}_t\|} \geq B_{N_s} \frac{\sum_{i=1}^{N_s} \beta_i \|\hat{\mathbf{x}}_i\|}{\|\mathbf{x}_\delta\|}. \tag{9}$$

Additionally, from Equation (5), we know that:

$$\|\mathbf{x}_\delta\| = \left\|\sum_{i=1}^{N_s} \beta_i \hat{\mathbf{x}}_i\right\|. \tag{10}$$

Thus, Equation (9) can be further represented as:

$$\tau\left(\mathbf{x}_\delta, \mathbf{y}_t\right) \ge B_{N_s} \frac{\sum_{i=1}^{N_s} \beta_i \|\hat{\mathbf{x}}_i\|}{\left\|\sum_{i=1}^{N_s} \beta_i \hat{\mathbf{x}}_i\right\|}. \tag{11}$$

According to the triangle inequality in vector spaces, we have $\left\|\sum_{i=1}^{N_s} \beta_i \hat{\mathbf{x}}_i\right\| \le \sum_{i=1}^{N_s} \beta_i \|\hat{\mathbf{x}}_i\|$. Therefore, we derive:

$$\tau\left(\mathbf{x}_\delta, \mathbf{y}_t\right) \ge B_{N_s} \cdot m, \quad \text{where } m \ge 1. \tag{12}$$

This result demonstrates that when effective source-modal proxies maintain a high cosine similarity with the true target, the adversarial example $\mathbf{x}_\delta$ will also achieve a high cosine similarity with the true target after optimization.

### D.2 THEOREM 3 FOR TARGET-MODAL PROXIES

For target-modal proxies, we can derive a similar theorem under slightly adjusted conditions:

**Theorem 3.** *Let $\mathbf{x}_\delta$ be the adversarial example generated by using multiple source-modal proxies $\{\hat{\mathbf{x}}_i\}_{i=1}^{N_s}$ and target-modal proxies $\{\hat{\mathbf{y}}_j\}_{j=1}^{N_c}$. Let us denote by $B_{N_c}$ the empirical lower bound of the cosine similarity between the adversarial example and the target-modal proxies, i.e., $B_{N_c} = \min_{j \in [N_c]} \tau\left(f_{\theta_S}(\mathbf{x}_\delta), f_{\theta_T}(\hat{\mathbf{y}}_j)\right)$. If $\mathbf{y}_t$ is an interior point of the convex polytope formed by the target-modal proxies $\{\hat{\mathbf{y}}_j\}_{j=1}^{N_c}$, then the similarity between the adversarial example and the true target will be lower bounded by $B_{N_c}$, i.e.,*

$$\tau\left(f_{\theta_S}(\mathbf{x}_\delta), f_{\theta_T}(\mathbf{y}_t)\right) \ge B_{N_c}.$$

The proof of Theorem 3 follows the same steps as Theorem 2. Theorem 3 implies that, if we set effective and comprehensive target-modal proxies such that the convex polytope encloses the true target, the adversarial example $\mathbf{x}_\delta$ will generalize to the true target. This result highlights the importance of designing high-quality proxies to improve the performance of AEs.

Combining Theorems 2 and 3, we can conclude that improving the effectiveness of the proxy targets (source-modal or target-modal) can enhance the generalizability of AEs.

## E IMPLEMENTATION DETAILS

**Overview of classification settings:** In this task, our goal is to determine whether the AEs could be classified as the target class when the true targets are used as classification prompts. Here, the estimated distribution $\mathcal{P}_{\text{target}}(\mathbf{Y} \sim \mathcal{D}_{\text{T}}|Q)$ which contains true targets are constructed by different text descriptions representing the same entity class. These descriptions are generated using various methods, including manually designed templates with varying styles and descriptions produced by Large Language Models. We evaluate the performance of AEs by measuring their Classification ASR (*Cls ASR*) against true prompts.

**Overview of retrieval setting:** In this task, the objective is to determine if AEs could align more strongly with true targets than benign examples, thus achieving effective targeted attacks. Here, $\mathcal{P}_{\text{target}}(\mathbf{Y} \sim \mathcal{D}_{\text{T}}|Q)$ corresponds to a scene with multiple entity classes, and we measure the success rate of retrieving AEs within this target scene. For example, in text-to-image retrieval, the attacker may only know a single keyword in the true target, like "dog", which results in a high-variance estimated distribution, or three keywords, such as "dog", "person", and "boat", leading to a lower-variance distribution. Adversarial performance is evaluated using *R@1 ASR* against the true target.

## E.1 Model, Task and Dataset Settings

We evaluate our method on both classification and retrieval tasks. The experimental setup for each dataset is detailed below. For all datasets, we pre-select specific entity classes or class combinations as base and target classes for generating AEs. The base classes include potentially harmful categories (e.g., firearms and explosives) to simulate real-world adversarial scenarios.

### E.1.1 Detailed of Experimented Models

**ImageBind.** ImageBind (Girdhar et al., 2023) learns a single shared embedding space by using images as the central hub to align heterogeneous modalities. It supports six modalities (image, text, audio, depth, thermal, IMU) and trains modality-specific encoders with CLIP-style contrastive objectives so that each non-image modality is bound to the image space via available pairwise datasets (e.g., image-text, image-audio), without requiring all modalities to be co-observed. Architecturally, it employs modality encoders with projection heads into a common d-dimensional space. This design enables zero-shot cross-modal retrieval and classification, including transfer between modality pairs that were never directly paired during training, while performance depends on the quality and coverage of image-centric pairs and remains encoder-only (non-generative).

**LanguageBind.** LanguageBind (Zhu et al., 2024) uses language as the pivot, mapping multiple modalities into a language-aligned embedding space so that text serves as a universal interface for cross-modal retrieval and zero-shot classification. It typically adapts modality encoders to align with a strong text encoder using contrastive learning on text-image, text-audio, and related pairs, sometimes adding lightweight adapters to preserve upstream priors. The approach is compatible with prompt engineering and instruction-tuned language models, often improving interoperability when labels, queries, or control signals are textual.

**One-PEACE.** One-PEACE (Wang et al., 2023a) provides a unified pretraining framework for image, text, and audio within a single backbone, combining discriminative alignment (contrastive) with representation objectives (masked/sequence modeling). A shared transformer with modality-aware embeddings and projection heads supports both unimodal and cross-modal tasks, offering a compact alternative to separate encoders while covering three major modalities. This unified design yields competitive zero-shot retrieval and classification across the supported modalities, though it covers fewer modalities than image- or language-pivot models and requires careful objective balancing to prevent any single modality from dominating capacity.

### E.1.2 Attack Settings for Retrieval Tasks

We perform text-to-image and audio-to-image retrieval tasks using the MSCOCO and XmediaNet datasets. Adversarial settings simulate varying levels of prior knowledge about the user's query.

***MSCOCO:*** The MSCOCO dataset provides extensive image-text pairs, making it suitable for text-to-image retrieval tasks. In this task, text descriptions serve as user queries, and the images act as retrieval targets.

**Text-to-image retrieval:**

- **Knowledge of Adversary:** The adversary is assumed to know specific keywords from the user's query, such as "["car", "person", "boat"]", "["boat", "person"]", "["person ', "bird"]", or "["boat"]". These are entity categories that occurred in the true queries.

- **Setup:** For each keyword combination, we extract text captions from MSCOCO that include the keywords. These captions are divided into two disjoint sets:
    1. **Target-Modal Proxies:** Text samples approximating the user's input, representing samples drawn from the estimated distribution $\mathcal{P}_{\text{target}}(\mathbf{Y} \sim \mathcal{D}_{\text{T}}|Q)$.
    2. **True Queries:** Representing the actual text input by the user.

- **Source-Modal Proxies:** Corresponding images associated with target-modal proxies serve as source-modal proxies sampled from $\mathcal{P}_{\text{target}}(\mathbf{X} \sim \mathcal{D}_{\text{S}}|Q)$.

***XmediaNet:*** XmediaNet is a multimodal dataset labeled by categories, enabling both cross-modal retrieval and classification tasks. We perform both text-to-image and audio-to-image retrieval tasks on XmediaNet.

1. **Text-to-Image Retrieval:**

   - **Knowledge of Adversary:** The adversary knows the category of the user's query, such as "airplane", "bear", "bomb", and "rifle" but does not have access to the full query sentences.
   - **Setup:** Text descriptions for each category are partitioned into two disjoint sets:
     (a) **Target-Modal Proxies:** Text approximations of the user's query, representing samples drawn from $\mathcal{P}_{\text{target}}(\mathbf{Y} \sim \mathcal{D}_{\text{T}}|Q)$.
     (b) **True Queries:** Representing the actual text input by the user.
   - **Source-Modal Proxies:** Corresponding images belonged the known category serve as source-modal proxies sampled from $\mathcal{P}_{\text{target}}(\mathbf{X} \sim \mathcal{D}_{\text{S}}|Q)$.

   **Audio-to-image retrieval:**

2. 
   - **Knowledge of Adversary:** The adversary knows category-level information ("airplane", "bear", "bomb", and "rifle") but lacks access to the full query from the user (audio instance).
   - **Setup:** Audio instances belonging to each category (e.g., bomb explosion sound represents category "bomb") are partitioned into two disjoint sets:
     (a) **Target-Modal Proxies:** Audio approximations of the user's query, representing samples drawn from $\mathcal{P}_{\text{target}}(\mathbf{Y} \sim \mathcal{D}_{\text{T}}|Q)$.
     (b) **True Queries:** Representing the actual audio input by the user.
   - **Source-Modal Proxies:** Corresponding images belonged the known category serve as source-modal proxies sampled from $\mathcal{P}_{\text{target}}(\mathbf{X} \sim \mathcal{D}_{\text{S}}|Q)$.

### E.1.3 ATTACK SETTINGS FOR CLASSIFICATION TASKS

We conduct zero-shot classification tasks using ImageNet and XmediaNet datasets. In these tasks, we assume the adversary has some knowledge about the user's classification prompt, such as specific categories, but lacks detailed information about the exact prompts.

***ImageNet:*** ImageNet, a widely used dataset for image classification, consists of 1000 categories. We perform zero-shot classification using text as prompts.

**Zero-shot classification (text as prompts):**

- **Adversarial Knowledge:** The adversary is aware of the categories ( "Shetland Sheepdog", "tree frog", "cannon", "rifle") of the prompts but lacks detailed knowledge about the user's exact prompts.

- **Target-Modal Proxies:** Text prompts representing various descriptions of the same category, synthesized using handcrafted prompt templates (Radford et al., 2021) and LLM-generated descriptions (Menon & Vondrick, 2022), are used as proxies sampled from the estimated distribution $\mathcal{P}_{\text{target}}(\mathbf{Y} \sim \mathcal{D}_{\text{T}}|Q)$.

- **True Prompt:** Additional text descriptions for the same category, generated by LLM (Menon & Vondrick, 2022), serve as the user's true input.

- **Source-Modal Proxies:** Image instances corresponding to the same category serve as proxies sampled from $\mathcal{P}_{\text{target}}(\mathbf{X} \sim \mathcal{D}_{\text{S}}|Q)$.

***XmediaNet:*** XmediaNet is a multimodal dataset comprising 200 categories, with each category containing text, image, and audio samples. We evaluate both text and audio prompts for zero-shot classification tasks.

1. **Zero-Shot Classification (Text as Prompts):**

- **Adversarial Knowledge:** The adversary knows the categories ("airplane", "bear", "bomb", "rifle") of user prompts but does not have access to the exact classification prompts from the user.
- **Target-Modal Proxies:** Text prompts generated using handcrafted templates (Radford et al., 2021), representing different descriptions for the same category, serve as proxies sampled from $\mathcal{P}_{\text{target}}(\mathbf{Y} \sim \mathcal{D}_{\text{T}}|Q)$.
- **True Prompt:** A generic prompt "a photo of a {class}", serves as the user's true input.
- **Source-Modal Proxies:** Image instances corresponding to the same category are used as proxies sampled from $\mathcal{P}_{\text{target}}(\mathbf{X} \sim \mathcal{D}_{\text{S}}|Q)$.

2. **Zero-Shot Classification (Audio as Prompts):**

- **Adversarial Knowledge:** The adversary knows category-level information of the prompt ("airplane", "bear", "bomb", "rifle") but lacks access to the exact audio prompt from the user.
- **Target-Modal Proxies:** Audio instances from the category serve as proxies sampled from $\mathcal{P}_{\text{target}}(\mathbf{Y} \sim \mathcal{D}_{\text{T}}|Q)$.
- **True Prompt:** Another disjoint set of audio instances from the category is used as the user's true input.
- **Source-Modal Proxies:** Image instances corresponding to the same category are used as proxies sampled from $\mathcal{P}_{\text{target}}(\mathbf{X} \sim \mathcal{D}_{\text{S}}|Q)$.

### E.2 COMPARED BASELINES

**Illusion Attack (Zhang et al., 2024b).** This targeted, cross-modal attack perturbs a source (e.g., image or audio) so that its embedding closely matches an adversary-chosen target in *another* modality (e.g., text), thereby "hallucinating" the target semantics in a shared embedding space. The optimization is cosine-similarity based and model-agnostic (works with CLIP-like encoders and other multi-modal embedding models). Empirically, it is highly effective when the evaluation target matches the optimization target, but generalization to unseen targets can drop.

**CrossFire (Dou et al., 2024).** CrossFire addresses cross-modal mismatch by first *converting* the attacker-chosen target into the *same* modality as the source (e.g., render the target text into an image/audio surrogate by generative models), then minimizing the L2 distance between the converted normalized target embedding and the normalized perturbed source embedding. This "modality matching" prior often improves optimization stability relative to directly chasing a cross-modal target and is instantiated as a cosine/angle minimization problem.

**MF-ii (Zhao et al., 2024b).** MF-ii (multi-facet, image-image) is a targeted transfer attack widely used in VLM robustness studies: given a target *text*, it first synthesizes a *target image* (e.g., via diffusion or find an image in the public dataset) conditioned on that text, then crafts an adversarial image by minimizing the cosine feature distance to the generated target image, thus turning the cross-modal objective into an image-image matching problem that transfers across VLMs.

**Sep-Attack (Madry et al., 2019; Li et al., 2020b).** Sep-Attack is a strong untargeted baseline that *independently* perturbs each modality without cross-modal coupling: PGD (or MI-PGD) is applied on images and BERT-Attack (token substitutions under semantic/fluency constraints) on texts; the two are then combined for multi-modal tasks. It is simple, scalable, and transferable, but typically underuses cross-modal interactions compared with later multimodal-coordinated methods.

**Co-Attack (Zhang et al., 2022).** Co-Attack perturbs *both* image and text jointly with explicit cross-modal coupling so that gradients and constraints reflect alignment behavior in VLP models (e.g., ALBEF, TCL). This coordinated optimization improves white-box effectiveness and can boost transfer over purely separate attacks, serving as a common multimodal baseline in retrieval and VE tasks. Open-source code is available.

**SGA (Lu et al., 2023).** SGA targets *adversarial transferability* by leveraging *set-level* cross-modal interactions and alignment-preserving augmentations. Instead of optimizing against a single pair, SGA aligns a *set* of text-image pairs to better capture many-to-many multimodal correspondences in VLPs, substantially improving black-box transfer on image-text retrieval benchmarks over Sep-/Co-Attack.

**CMI-Attack (Fu et al., 2024).** CMI-Attack explicitly exploits modality interactions during optimization, e.g., using *embedding-level* text perturbations that preserve semantics and *interaction-guided* image gradients to constrain both modalities. This yields stronger cross-model transfer and improved cross-task generalization in vision-language retrieval relative to prior baselines.

We follow the original settings of each baseline whenever possible, unless otherwise noted.

### E.3 EVALUATION METRICS

#### E.3.1 CLASSIFICATION TASKS

We evaluate attack effectiveness using the *Classification Attack Success Rate* (*Cls ASR, %*), defined as:

$$Cls\ ASR\ (\%) = \frac{|A_{\text{success}} \setminus A'_{\text{success}}|}{N_{\text{total}}} \times 100, \tag{13}$$

where $A_{\text{success}}$ is the set of AEs classified as the target class after the attack, $A'_{\text{success}}$ is the set already classified as the target class before the attack, and $N_{\text{total}}$ is the total number of generated AEs. A higher ASR indicates a more effective attack.

When anomaly detection is enabled, let $A_{\text{detected}} \subseteq A_{\text{success}} \setminus A'_{\text{success}}$ denote the subset of successful AEs that are detected. The *Classification Attack Success Rate after anomaly Detection* (*Cls ASRD, %*) is:

$$Cls\ ASRD\ (\%) = \frac{|(A_{\text{success}} \setminus A'_{\text{success}}) \setminus A_{\text{detected}}|}{N_{\text{total}}} \times 100, \tag{14}$$

so a higher ASRD indicates greater effectiveness in the presence of anomaly detection.

#### E.3.2 RETRIEVAL TASKS

We measure performance using the *Recall@K Attack Success Rate* (*R@K ASR, %*), defined as:

$$R@K\ ASR\ (\%) = \frac{|A_{\text{success}} \setminus A'_{\text{success}}|}{N_{\text{test}}} \times 100, \tag{15}$$

where $A_{\text{success}}$ is the set of test queries for which injected AEs are retrieved within rank $K$ after the attack, $A'_{\text{success}}$ is the set already retrieved within rank $K$ before the attack, and $N_{\text{test}}$ is the total number of test queries.

With anomaly detection, let $A_{\text{detected}} \subseteq A_{\text{success}} \setminus A'_{\text{success}}$ be the set of successful yet detected cases. The *Recall@K after anomaly Detection* (*R@K ASRD, %*) is:

$$R@K\ ASRD\ (\%) = \frac{|(A_{\text{success}} \setminus A'_{\text{success}}) \setminus A_{\text{detected}}|}{N_{\text{test}}} \times 100, \tag{16}$$

where a higher ASRD indicates stronger attack effectiveness under anomaly detection.

#### E.3.3 ANOMALY DETECTION SETTINGS

For anomaly detection, we use the same datasets and task configurations as those in the attack evaluations to assess the detection performance of AEs generated in these scenarios. The detection framework focuses on analyzing the top-$K$ samples retrieved by the model, identifying the most suspicious samples that may be adversarial. In addition to our proposed anomaly detection method, we compare its performance with some unsupervised anomaly detection techniques, including Isolation Forest (Liu et al., 2008), PCA (Hoffmann, 2007), and $k$NN (Angiulli & Pizzuti, 2002). These baseline methods are also applied to the top-$K$ samples for consistent evaluation. For implementation, we use the off-the-shelf functionality provided by the PyOD library (Zhao et al., 2019) for these techniques.

### E.4 HYPERPARAMETER SETTINGS

For our detection approach, the number of iterations is set to $T = 2$, as performance converges quickly. For the proposed attack method PTA, the number of source-modal proxies ($N_s$) is set to 20 for retrieval tasks and 25 for classification tasks, while the number of target-modal proxies ($N_c$) is

set to 50 for retrieval tasks and 10 for classification tasks, unless otherwise specified. The balancing factor $\alpha$ is chosen based on the task and defense scenario. When anomaly detection is applied as a defense, we focus on ASRD performance with $\alpha = 0.4$ for retrieval tasks and $\alpha = 1.0$ for classification tasks. In cases without anomaly detection, $\alpha$ is set to 0 to prioritize the generalizability of attack and focus on ASR performance.

### E.5 SETTINGS AND RESULTS FOR POTENTIAL DEFENSE

**TeCoA**: We adversarially trains CLIP ViT/B-32 (Radford et al., 2021) to attenuate the adversarial features using TeCoA (Mao et al., 2023). In the main paper we report the result when the adversarial budget of 16/255 in retrieval task of XmediaNet and extended results are provided in Table 8.

Table 8: Impact of adversarially fine-tuned CLIP VIT/B-32 on *Cls ASR (%)* and *R@1 ASR (%)* performance of attacks for classification and retrieval tasks.

| | | Classification Task | | Retrieval Task | |
|---|---|---|---|---|---|
| **Attack** | $\epsilon$ | **XmediaNet** | **ImageNet** | **XmediaNet** | **MSCOCO** |
| Illusion Attack | 8/255 | $22.08_{0.59}$ | $10.57_{0.13}$ | $9.72_{0.36}$ | $3.94_{0.04}$ |
| **PTA (Ours)** | 8/255 | $\mathbf{22.50}_{0.00}$ | $\mathbf{13.73}_{0.05}$ | $\mathbf{14.20}_{0.35}$ | $\mathbf{4.64}_{0.07}$ |
| Illusion Attack | 16/255 | $80.42_{0.59}$ | $44.74_{0.01}$ | $62.31_{0.24}$ | $10.89_{0.02}$ |
| **PTA (Ours)** | 16/255 | $\mathbf{78.16}_{0.00}$ | $\mathbf{51.56}_{0.02}$ | $\mathbf{78.03}_{0.20}$ | $\mathbf{24.30}_{0.15}$ |
| Illusion Attack | 32/255 | $99.58_{0.59}$ | $68.72_{0.09}$ | $87.52_{0.00}$ | $22.55_{0.29}$ |
| **PTA (Ours)** | 32/255 | $\mathbf{100.00}_{0.00}$ | $\mathbf{74.00}_{0.16}$ | $\mathbf{97.78}_{0.08}$ | $\mathbf{64.42}_{0.11}$ |

**Data augmentation**: Following Zhang et al. (2024b), we use data augmentations of Gaussian Blur, JPEG, and Random Affine to disrupt adversarial features of AEs generated for LanguageBind on XmediaNet. Specifically, we optimize the adversarial noise by integrating differentiable approximations of these transformations and using Kornia (Shi et al., 2020) to compute gradients during backpropagation. We report the result of GaussianBlur of retrieval tasks in the main paper and extended results are provided in Table 9.

Table 9: Impact of **data augmentation** on *R@1 ASR (%)* for the retrieval task on XmediaNet with LanguageBind.

| | Retrieval Task | | |
|---|---|---|---|
| **Method** | **GaussianBlur** | **JPEG** | **RandomAffine** |
| Illusion Attack | $12.44_{0.13}$ | $9.87_{1.12}$ | $11.00_{0.48}$ |
| **PTA (Ours)** | $\mathbf{89.33}_{0.31}$ | $\mathbf{52.19}_{2.24}$ | $\mathbf{63.05}_{0.16}$ |

**DiffPure**: The diffusion-based purification (Nie et al., 2022) is used against AEs for LanguageBind in XmediaNet.. Because purification can introduce non-differentiability and stochasticity, we also test an adaptive attack using **BPDA+EOT** (Hill et al., 2021) to avoid gradient masking. We report the result of result of retrieval task in the main paper and extended results are provided in Table 10.

Table 10: Impact of **DiffPure** on *R@1 ASR (%)* for retrieval task on XmediaNet with LanguageBind.

| **Method** | **Classification Task** | **Retrieval Task** |
|---|---|---|
| Illusion Attack | $10.31_{0.09}$ | $9.83_{0.07}$ |
| **PTA (Ours)** | $\mathbf{67.62}_{0.12}$ | $\mathbf{71.97}_{0.37}$ |

## F ADDITIONAL EXPERIMENTS

### F.1 EXPLANATION OF THE VULNERABILITY OF CLASSIFICATION SYSTEM

From the main content's Tables 2 and 3, ASR/ASRD in classification are consistently higher than in retrieval. We hypothesize this is because, in classification, the (estimated) target distribution is more *concentrated* (class prompts), whereas retrieval involves more variable scenes and queries, yielding a more *dispersed* distribution. To support this, we compute the mean cosine similarity among samples

drawn from the estimated source/target distributions and the trace of their covariance matrices, $\text{tr}(\boldsymbol{\Sigma}_\text{T})$ and $\text{tr}(\boldsymbol{\Sigma}_\text{S})$ for both tasks, shown in Table 11. Retrieval exhibits markedly lower mean cosine similarity for both source- and target-modal proxies and higher covariance traces, indicating a broader spread. Thus, the estimated distributions $\mathcal{P}_\text{target}(\mathbf{X} \sim \mathcal{D}_\text{S} \mid Q)$ and $\mathcal{P}_\text{target}(\mathbf{Y} \sim \mathcal{D}_\text{T} \mid Q)$ are more dispersed in retrieval. This explains why the retrieval system is more robust with generalized AEs.

Table 11: Comparison of the mean cosine similarity and the trace of covariance matrix between the estimated target-modal and source-modal distribution for classification and retrieval.

| Task Type | Cosine Similarity | | Trace of Cov. Matrix | |
|---|---|---|---|---|
| | Target-Modal | Source-Modal | $\sigma_\text{T}$ | $\sigma_\text{S}$ |
| Classification | 0.7782 | 0.7247 | 0.1107 | 0.2753 |
| Retrieval | 0.3983 | 0.4246 | 0.5868 | 0.5706 |

**Varying prompt variety.** Building on this observation, we vary the diversity of $\mathcal{P}_\text{target}(\mathbf{Y} \sim \mathcal{D}_\text{T} \mid Q)$ by using different prompt templates for ImageNet class prompts:

- **Standard:** "a photo of a {class}."
- **Waffle:** Prompts with random words/characters (Roth et al., 2023).
- **Manual:** 80 manually curated generic prompts (Radford et al., 2021), e.g., "a drawing of the {class}."

Table 12 reports *Cls ASR (%)* for AEs trained/tested under different prompt sets. When test prompts are more diverse (e.g., Manual), ASR drops, supporting our conjecture and suggesting the defense of increase prompt variety to reduce target concentration and hinder generalized targeted attacks.

Table 12: Comparison of *Cls ASR (%)* across different train/test prompts with $\epsilon = 4/255$ on ImageNet.

| Train \Test | Standard | Waffle | Manual |
|---|---|---|---|
| Standard | 99.58 | 95.18 | 80.51 |
| Waffle | 99.58 | 95.28 | 79.95 |
| Manual | 99.58 | 95.48 | 83.57 |

### F.2 PTA's Effectiveness with Limited Adversarial Prior Knowledge

In retrieval, constructing the estimated true target distribution $\mathcal{P}_\text{target}(\mathbf{Y} \sim \mathcal{D}_\text{T} \mid Q)$ depends on prior knowledge (e.g., known entity keywords in the user query). We assess PTA's effectiveness with limited prior knowledge to different extents by varying how many entity classes are known to the attacker: one keyword (low prior), two (medium), and three (higher) on MSCOCO.

Table 13 shows that PTA maintains high success even with a single known keyword ("boat"), and its performance scales gracefully as prior knowledge increases, whereas Illusion Attack stays low at one/two keywords and only rises at three keywords. This demonstrates that PTA is effective under limited prior knowledge and improves further as knowledge grows.

### F.3 PTA's Effectiveness with Audio Target Modality

We further test PTA when the target shifts from discrete *text* to continuous *audio* on XmediaNet, evaluating audio-target generalizability for retrieval (R@1 aud ASR) and classification (Cls aud ASR) under the same protocol (Table 15. Additional settings are provided in Appendix E.1).

Across all three models and two tasks, PTA substantially exceeds the baseline. These results indicate that PTA does not depend on text-specific discretization effects. Rather, its proxy-driven objective transfers to continuous targets, where increased proxy diversity continues to provide versatile training signals and yields robust cross-target success without task-specific retuning.

Table 13: *R@1 ASR (%)* under varying amounts of prior knowledge (number of known entity classes) for Illusion Attack vs. PTA on MSCOCO.

| Known Ent. | Attack | ImageBind | LanguageBind | One-PEACE |
|---|---|---|---|---|
| "boat" | Illusion Attack | $12.50_{0.88}$ | $17.38_{0.53}$ | $6.38_{0.23}$ |
| | **PTA (Ours)** | $69.94_{0.97}$ | $89.04_{2.21}$ | $23.31_{0.09}$ |
| "boat", "person" | Illusion Attack | $12.88_{0.88}$ | $21.75_{0.00}$ | $6.06_{0.43}$ |
| | **PTA (Ours)** | $68.62_{0.88}$ | $88.25_{1.77}$ | $25.38_{0.53}$ |
| "boat", "person", "car" | Illusion Attack | $34.94_{1.86}$ | $49.00_{1.52}$ | $19.88_{0.35}$ |
| | **PTA (Ours)** | $80.44_{0.80}$ | $88.00_{0.52}$ | $53.12_{0.33}$ |

Table 14: Comparison results of **unknown-modal generalizability**. We report audio modality attack success rates (*Cls aud ASR (%)* and *R@1 aud ASR (%)*) on XmediaNet.

| Task | Method | ImageBind | LanguageBind | One-PEACE |
|---|---|---|---|---|
| Classification | Illusion Attack | $8.78_{0.00}$ | $13.13_{0.00}$ | $13.67_{0.00}$ |
| | **PTA (Ours)** | $\mathbf{11.08}_{0.06}$ | $\mathbf{17.00}_{0.39}$ | $\mathbf{17.64}_{0.05}$ |
| Retrieval | Illusion Attack | $9.11_{0.00}$ | $4.55_{0.00}$ | $13.65_{0.00}$ |
| | **PTA (Ours)** | $\mathbf{9.33}_{0.11}$ | $\mathbf{39.75}_{0.44}$ | $\mathbf{19.65}_{0.07}$ |

### F.4 PTA'S EFFECTIVENESS WITH UNKNOWN TARGET MODALITY

Beyond the generalizability paradigm discussed in the main text, we further explore a more challenging scenario: *unknown-modal generalizability*. In real-world cross-modal matching tasks, models often accept inputs from multiple modalities, meaning the adversary may not know the specific modality of the user's input. In such cases, we denote $\mathcal{P}_{\text{target}}^{\text{UM}}(\mathbf{U}|Q)$ as the distribution of potential targets constructed by the adversary when the target modality is unknown. The generalizability of AEs to unknown-modal targets is thus defined as:

$$\mathbf{G}_{\text{UM}}(\mathbf{x}_\delta) = \mathbb{E}_{\mathbf{u} \sim \mathcal{P}_{\text{target}}^{\text{UM}}(\mathbf{U}|Q)} \left[ \tau(f_{\theta_{\text{S}}}(\mathbf{x}_\delta), f_{\theta_{\text{U}}}(\mathbf{u})) \right],$$

where $f_{\theta_{\text{U}}}$ represents the encoder of the unknown modality, and $\mathbf{u}$ is a sample drawn from the estimated distribution of the unknown modality. For example, adversarial images created for text-to-image retrieval tasks may generalize to unknown modalities to the adversary, enabling them to function across various multimodal tasks, such as audio-to-image or image-to-image retrieval, even when the adversary lacks data from the unknown modality.

To evaluate the unknown-modal generalizability, we select audio modality as the adversary's unknown target modality, assessing the attack performance on audio-to-image retrieval and zero-shot classification (audio as prompt). We assess AE performance when $\mathcal{P}_{\text{target}}^{\text{UM}}(\mathbf{U}|Q)$ is the estimated distribution constructed by the adversary. In this scenario, we assume that the adversary only has access to text and image modalities for optimization and lacks information about the audio modality. We then test unknown-modal generalizability by evaluating performance on audio modality, using the XmediaNet dataset for evaluation.

- **Classification setting:** In this task, the true targets consist of audio samples that represent the semantically identical entity (e.g., dog barking sounds), serving as audio prompts for image classification. In this scenario, the adversary has no access to audio data and relies only on text and image data as proxy targets to generate AEs.

- **Retrieval setting:** In this task, the true targets are also audio samples. Similarly, the adversary has access only to text and image data. We evaluate R@1 ASR in audio-to-image retrieval tasks to determine if AEs achieve effective targeted attacks.

In Table 14, we illustrate the ASR for audio modality for both classification and retrieval. PTA outperforms the Illusion Attack, which aligns with a single target in the text modality, in terms of unknown-modal generalizability. These findings suggest that using multiple proxy targets and source-modal optimization improves the generalizability of AEs across previously unseen modalities.

Table 15: Comparison results of audio-target-modal generalizability. We report ASR (*Cls aud ASR (%)* and *R@1 aud ASR (%)*) when the target modality is audio on the XmediaNet.

| Task | Method | ImageBind | LanguageBind | One-PEACE |
|---|---|---|---|---|
| Classification | Illusion Attack | $30.46_{0.00}$ | $50.92_{0.00}$ | $34.97_{0.00}$ |
| | **PTA (Ours)** | $\textbf{53.08}_{0.31}$ | $\textbf{73.11}_{0.67}$ | $\textbf{51.24}_{0.78}$ |
| Retrieval | Illusion Attack | $40.51_{0.00}$ | $58.56_{0.00}$ | $42.37_{0.00}$ |
| | **PTA (Ours)** | $\textbf{65.37}_{0.41}$ | $\textbf{89.34}_{0.88}$ | $\textbf{59.81}_{0.55}$ |

## F.5 ADDITIONAL COST OF PTA

PTA introduces one extra component beyond normal adversarial attacks: a set of proxy embeddings. Crucially, proxy collection and embedding computation are performed *offline* on high-performance machines before any attack is executed. At attack time, the optimization only consumes a few additional lookups/inner-products against the precomputed proxies, so the online overhead is negligible. In specific, in *classification*, AE optimization runs on the attacker's device, while proxy collection still happens offline without time constraints. In *retrieval*, the adversary collects proxies and optimizes adversarial examples (AEs) offline (e.g., using generative models, public datasets, or web sources), then uploads the finalized AEs to the gallery. All heavy computation occurs off device, so the low-resource client does not run the optimization procedure. Since PTA only adds precomputed proxy embeddings during optimization, the extra compute/memory cost is minimal.

As shown in Table 16, for ImageBind on an NVIDIA V100 with 100 target proxies and 50 source proxies, PTA incurs only a *0.16%* increase in optimization time per epoch and a *0.03%* increase in GPU memory versus a normal adversarial attack (PGD). These differences are practically negligible, confirming that PTA remains suitable for low-compute settings.

Table 16: Compute overhead of PTA vs. a vanilla adversarial attack (no proxies). Numbers are measured on ImageBind with 100 target proxies and 50 source proxies.

| Method | Optimization time per epoch | GPU memory used |
|---|---|---|
| Normal adversarial attack | 121.1 ms | 6184 MB |
| **PTA (Ours)** | 121.3 ms (↑ 0.16%) | 6186 MB (↑ 0.03%) |

# G USE OF LLMS

We used LLMs solely as writing assistants for *language refinement*. Concretely, LLM prompts were limited to grammar correction, style tightening, phrasing alternatives, and minor re-organization of paragraphs for clarity and brevity. All LLM-suggested edits were reviewed and verified by the authors, and all technical content is author-generated and author-validated.

