# OpenReview forum: "Improving Generalizability and Undetectability for Targeted Adversarial Attacks on Multimodal Pre-trained Models"
_ICLR.cc/2026/Conference — ICLR 2026 Conference Withdrawn Submission_

### Official Review · Reviewer_c88u · 2025-10-23

**Soundness:** 2
**Presentation:** 2
**Contribution:** 2
**Rating:** 4
**Confidence:** 4

**Summary:**

This paper investigates the security vulnerabilities of large-scale multimodal pre-trained models to targeted adversarial attacks, with a focus on cross-modal matching tasks. The authors identify two major unresolved issues in existing methods: poor generalizability (i.e., adversarial examples only work on an exact known target and fail on semantically similar but different targets) and limited undetectability (i.e., adversarial examples can be easily flagged as outliers by simple anomaly detection). To address these, they introduce the Proxy Targeted Attack (PTA), which leverages both source-modal and target-modal proxies in its optimization. Theoretical analysis is provided to formalize a trade-off between generalizability and undetectability, and comprehensive experiments on several state-of-the-art multimodal models and tasks show improvements over prior approaches in attack success, generalization, and stealthiness.

**Strengths:**

**Comprehensive ablations**: The roles of key hyperparameters are dissected in Figure 4 and Figure 5, providing practical insights for real use.

**Broadened evaluation**: The method is also tested in black-box settings, on text and audio modalities, and remains potent—even outperforming strong baselines.

**Weaknesses:**

**Limited conceptual novelty**：Although the proposed Proxy Targeted Attack (PTA) framework is presented in a novel form, its core idea—leveraging proxy samples to enhance the generalizability and stealthiness of adversarial examples—is essentially an incremental extension of existing targeted attack paradigms. The main contributions lie in the combination of loss components and the introduction of proxy sets, which, while systematically analyzed both theoretically and empirically, offer limited conceptual or methodological innovation.

**Placement and accessibility of key details**：Many important methodological and theoretical clarifications—such as the proxy selection strategy, detection metric assumptions, and additional defense results—are included in the appendix rather than the main body. While this is understandable given space constraints, some of these points (e.g., proxy sampling and update policy, sensitivity analysis to detection thresholds) are central to understanding the method. Moving key portions into the main text or providing a concise summary section would substantially enhance readability for the reader.

**Limited theoretical validation beyond simplified assumptions**：Although the appendix provides further explanation on Theorem 1 and the trade-off formalization, the assumptions (notably the fixed anomaly threshold β and L₂-based distance metric) remain rather idealized.

**Defense coverage and interpretation**: The defense evaluation (Table 7 and Appendix E) includes adversarial training and data augmentation, but it still lacks certified or adaptive detection defenses such as MMCert. Additionally, while PTA maintains high success rates under existing defenses, the paper could benefit from a more detailed analysis of why. For instance, whether the proxy mechanism intrinsically avoids defense gradient masking or shifts feature-space distributions differently.

**Discussion of model-specific behavior**：In Table 4, One-PEACE exhibits smaller degradation compared to other multimodal models. The appendix briefly mentions architectural differences, but more explicit discussion in the main text would improve interpretability and generalization insights.

**Questions:**

1.Can the authors elaborate on how proxies are actually selected—random sampling, clustering, or some other heuristic?

2.How sensitive is the theoretical guarantee (Theorem 1, 2) to the specific choice of anomaly detection method and the distance metric?

3.Why were key certified defenses like MMCert not included in the defense benchmark (Table 7)? Are there practical or conceptual challenges preventing such experiments, or would PTA fundamentally fail under those defense strategies?

4.In Table 4, One-PEACE appears less affected by injection of PTA-crafted AEs than other models, especially at low injection rates. Can the authors offer an explanation—is this due to architectural differences, training diversity, or other factors? Is this generalizable to other multimodal architectures?

5.Could you detail the practical cost (runtime, memory) of optimizing PTA with large proxy sets across high-dimensional modalities?

---

### Official Review · Reviewer_TJtz · 2025-10-28

**Soundness:** 3
**Presentation:** 3
**Contribution:** 3
**Rating:** 6
**Confidence:** 4

**Summary:**

This paper investigates adversarial attacks that are both generalizable and undetectable, focusing on multimodal pre-trained models such as ImageBind.
The authors argue that existing attacks lack both properties, limiting their practical effectiveness.
To address this issue, the paper proposes a method called Proxy Targeted Attack (PTA), which leverages multiple source-modal and target-modal proxies to generate adversarial examples (AEs).
By using multiple target-modal proxies, AEs are optimized to align with a target-modal distribution, thereby improving generalization.
By using source-modal proxies, AEs are constrained to remain within the clean sample distribution, improving undetectability.
Furthermore, the paper provides a theoretical analysis showing that there exists a fundamental trade-off between the generalizability and undetectability of AEs.

**Strengths:**

- Attacking multimodal pre-trained models such as ImageBind and LanguageBind, beyond the typical vision-language setting, is an important and novel topic.
- Identifying the challenge of achieving both generalizability and undetectability in the multimodal settings, and demonstrating that both can be simultaneously improved through appropriate optimization, is valuable. The improvement in attack capability over baselines is significant.
- The proposed method is conceptually simple.
- The experiments are extensive, covering both image classification and retrieval tasks, and involving not only image-text but also text-audio modalities.

**Weaknesses:**

- The trade-off between adversarial attack strength and undetectability has already been discussed in prior work [Frederickson et al., 2018], and is also somewhat intuitive. Therefore, the theoretical conclusion in Section 2.4 has limited novelty.
- The idea of improving generalizability using multiple data points is not new; a similar approach was proposed in SGA-Attack [Lu et al., 2023]. The paper should acknowledge and discuss this connection.
- The idea of modifying the objective function to jointly optimize attack capability and undetectability was also discussed in [Frederickson et al., 2018], and this relationship should be mentioned for completeness.


[Frederickson+2018] Frederickson, Christopher, et al. "Attack strength vs. detectability dilemma in adversarial machine learning." 2018 international joint conference on neural networks (IJCNN). IEEE, 2018.

[Lu+2023] Lu, Dong, et al. "Set-level guidance attack: Boosting adversarial transferability of vision-language pre-training models." Proceedings of the IEEE/CVF International Conference on Computer Vision. 2023.

**Questions:**

see weaknesses

---

### Official Review · Reviewer_KZyb · 2025-10-30

**Soundness:** 3
**Presentation:** 3
**Contribution:** 2
**Rating:** 2
**Confidence:** 5

**Summary:**

This paper introduces a targeted adversarial attack against multimodal encoders, characterized by its generalizability and undetectability. The authors propose a Proxy Targeted Attack (PTA), which leverages multiple source- and target-modal proxies to optimize targeted adversarial examples (AEs), enabling them to evade defenses while aligning with multiple potential targets. The paper further provides theoretical analyses to elucidate the relationship between generalizability and undetectability, ensuring optimal generalizability under the specified undetectability constraints. Experimental results demonstrate the effectiveness of the proposed method.

**Strengths:**

1、The motivation of the paper is clearly articulated, making the overall narrative easy to follow.

2、The authors provide theoretical analyses to highlight the relationship between generalizability and undetectability.

3、The release of open-source code ensures the reproducibility of the work.

**Weaknesses:**

1、The technical novelty of this work is quite limited. The proposed method essentially aligns the adversarial example’s features with multiple injected features— a strategy that has become rather common in recent multimodal attack studies. For instance, SGA [1] (ICCV 2023) and AttackVLM [2] (NeurIPS 2023) both adopt similar attack paradigms. Notably, the paper incorrectly cites AttackVLM as a NeurIPS 2024 paper. The only notable difference here is that the proposed method introduces benign image features as constraints to enhance undetectability, which is also a widely adopted practice in adversarial attack design. In fact, many recent works have already begun exploring stealthiness in digital, feature, and frequency domains. Therefore, I find it difficult to see how this work provides any substantial advancement to the current state of multimodal adversarial attack research.

2、The adversarial detection baselines considered in this paper are quite outdated, mostly predating 2009. How does the proposed method perform against more recent and advanced adversarial detection techniques, such as frequency-based methods [3,4,5], spatial-based methods [5,6], or feature-based methods [7]? I strongly recommend the authors review and discuss the latest progress summarized in the comprehensive survey [8], and conduct experiments comparing their approach with these more recent detection defenses. Such evaluation would significantly strengthen the paper’s contribution.

3、The black-box attack evaluation is weak: only one baseline from 2020 is considered, which is unconvincing. The authors should include more recent black-box target attack methods for comparison and discussion, such as [9,10,11].

4、In the defense evaluation section, the authors show that the proposed PTA method remains effective against defenses like TeCoA (adversarial training), Gaussian Blur (data augmentation), and DiffPure. However, the method is not explicitly designed to counter these defense mechanisms. Why does PTA still succeed under these settings? This is especially puzzling in the case of DiffPure, which is well known for its effectiveness in purifying adversarial noise. The authors should provide a more detailed explanation of these results and clarify the underlying mechanism that enables PTA to bypass such defenses.

5、The experimental evaluation mainly focuses on white-box settings. It remains unclear how well the generated adversarial samples transfer across different models.

6、The paper lacks a discussion and comparison with recent state-of-the-art multimodal adversarial attacks [12,13,14,15].


The paper lacks technical novelty, as it is largely based on existing work without substantial innovation. Moreover, the experiments are limited and outdated, lacking comparisons with recent adversarial detection methods, state-of-the-art multimodal attacks, and thorough black-box and transferability evaluations.

Reference

[1] Set-level Guidance Attack: Boosting Adversarial Transferability of Vision-Language Pre-training Models, ICCV 2023.

[2] On Evaluating Adversarial Robustness of  Large Vision-Language Models, NeurIPS 2023.

[3] Automated Detection System for Adversarial Examples with High-Frequency Noises Sieve, International Symposium on Cyberspace Safety and Security 2019.

[4] Detecting AutoAttack Perturbations in the Frequency Domain, ICML 2021 workshop.

[5] Feature Squeezing: Detecting Adversarial Examples in Deep Neural Networks, NDSS 2017.

[6] Adversarial Example Detection Using Latent Neighborhood Graph, ICCV 2021.

[7] Detecting Adversarial Examples through Image Transformation, AAAI 2018.

[8] Adversarial example detection for DNN models: a review and experimental comparison, Artificial Intelligence Review 2022.

[9] Improving Transferable Targeted Adversarial Attacks with Model Self-Enhancement, CVPR 2024.

[10] Boosting the Transferability of Adversarial Examples via Local Mixup and Adaptive Step Size, ICASSP 2025.

[11] Enhancing targeted transferability via feature space fine-tuning, ICASSP 2024.

[12] One perturbation is enough: On generating universal adversarial perturbations against vision-language pre-training models, CVPR 2025.

[13] Exploring transferability of multimodal adversarial samples for vision-language pre-training models with contrastive learning. TMM 2025.

[14] AnyAttack: Towards Large-scale Self-supervised Adversarial Attacks on Vision-language Models, CVPR 2025.

[15] Modality-Specific Interactive Attack for Vision-Language Pre-Training Models, TIFS 2025.

**Questions:**

It is also recommended that the authors include attack results on large multimodal models, such as LLaVA, MiniChatGPT, and GLM, to further validate the method’s generalization and practicality.

---

### Official Review · Reviewer_zTv8 · 2025-11-01

**Soundness:** 3
**Presentation:** 3
**Contribution:** 3
**Rating:** 4
**Confidence:** 3

**Summary:**

This paper proposes Proxy Targeted Attack (PTA), a method to improve both generalizability and undetectability of targeted adversarial examples on multimodal pre-trained models such as ImageBind and LanguageBind. PTA introduces source- and target-modal proxies to craft adversarial examples that remain stealthy while aligning with multiple semantically related targets. Theoretical analyses reveal the trade-off between the two goals, and extensive experiments across classification, retrieval, and multimodal settings show PTA’s superior attack success rate.

**Strengths:**

1. This paper proposes a novel and well-motivated approach addressing two key weaknesses in adversarial attacks on multimodal pretraining models.
2. This paper provides comprehensive experiments across models, modalities, and defense scenarios.
3. The paper is well organized, with clear problem statement and detailed method description.

**Weaknesses:**

1. Given that LLaVA employs a vision encoder (e.g., CLIP) pretrained on large-scale vision-language data, it would strengthen the paper to investigate whether the proposed attack can be effectively adapted to such popular large vision-language models (LVLMs).

2. The transferability across models of the proposed attack should be benchmarked against established baselines such as SGA [1] under the experimental conditions defined in the original SGA paper.

3. The practicality of PTA under fully blind attacks (no target prior) is not explored; the authors should quantify performance degradation.

4. The core idea of the proposed method relies on cross-modal interaction, which appears conceptually similar to prior work (e.g., SGA [1]). The authors should clearly articulate how their approach differs from existing methods.

5. The parameter α governs the trade-off between generalizability and undetectability; the authors should further clarify how α is selected and provide a detailed analysis of its impact on model performance.


[1] Lu D., Wang Z., Wang T., et al. Set-level guidance attack: Boosting adversarial transferability of vision-language pre-training models. ICCV 2023.

**Questions:**

Please address the weakness above.

---

### Note · Authors · 2025-11-23

I have read and agree with the venue's withdrawal policy on behalf of myself and my co-authors.